# Inhibiting adipose tissue M1 cytokine expression decreases DPP4 activity and insulin resistance in a type 2 diabetes mellitus mouse model

Lee-Wei Chen[1,2,3]*, Pei-Hsuan Chen[1], Jui-Hung Yen[4]*

1 Department of Surgery, Kaohsiung Veterans General Hospital, Kaohsiung, Taiwan, 2 Institute of Emergency and Critical Care Medicine, National Yang Ming Chiao Tung University, Taipei, Taiwan, 3 Department of Biological Sciences, National Sun Yat-Sen University, Kaohsiung, Taiwan, 4 Department of Microbiology and Immunology, Indiana University School of Medicine, Indianapolis, Indiana, United States of America

* lwchen@vghks.gov.tw (L-WC); jimyen@iu.edu (J-HY)

**Data Availability Statement:** All relevant data are within the paper and its Supporting Information files.

## Abstract

Adipose tissue inflammation is a major cause of the pathogenesis of obesity and comorbidities. To study the involvement of M1/M2 cytokine expression of adipose tissue in the regulatory mechanisms of dipeptidyl peptidase 4 (DPP4) and insulin resistance in diabetes, stromal vascular fractions (SVFs) were purified from inguinal adipose tissue of diabetic (*Lepr^db/db*) and non-diabetic (*Lepr^+/+*) mice followed by analysis of M1/M2 cytokine expression. SVFs of *Lepr^db/db* mice exhibited increased TNF-α, IL-6, IL-1β, CCL2, and DPP4 mRNA expression but decreased IL-10 mRNA expression. Plasma from *Lepr^db/db* mice induced TNF-α, IL-6, IL-1β, CCL2, and DPP4 mRNA expression and plasma from *Lepr^+/+* mice induced IL-10 mRNA expression in SVFs from *Lepr^db/db* mice. Injection of *Lepr^+/+* plasma into the adipose tissue of *Lepr^db/db* mice decreased mRNA expression of TNF-α, IL-6, IL-1β, CCL2, and DPP4 and protein expression of pJNK and DPP4 in SVFs, reduced mRNA expression of ICAM, FMO3, IL-1β, iNOS, TNF-α, IL-6, and DPP4 and protein expression of ICAM, FMO3, and DPP4 in liver, and suppressed mRNA expression of TNF-α, IL-6, IL-1β, and DPP4 in Kupffer cells. Plasma from *Lepr^db/db* mice did not induce M1 cytokine expression in SVFs from *Lepr^db/db*-*Jnk1^-/-* mice. Altogether, we demonstrate that diabetes induces M1 but decreases M2 cytokine expression in adipose tissue. Diabetic plasma-induced M1 expression is potentially through pJNK signaling pathways. Non-diabetic plasma reverses M1/M2 cytokine expression, plasma CCL2 levels, DPP4 activity, and Kupffer cell activation in diabetes. Our results suggest M1/M2 cytokine expression in adipose tissue is critical in diabetes-induced DPP4 activity, liver inflammation, and insulin resistance.

## Introduction

Dysregulation of M1/M2 polarization in adipose tissue is one of the major mechanisms underlying the pathogenesis of obesity and comorbidities, such as insulin resistance and

**Funding:** LW Chen: National Science Council (MOST 107-2314-B-010-043-MY3) www.most. gov.tw LW Chen: Kaohsiung Veterans General Hospital (VGHKS109-097) www.vghks.gov.tw The funders had no role in study design, data collection and analysis, decision to publish, or preparation of the manuscript.

**Competing interests:** The authors have declared that no competing interests exist.

nonalcoholic fatty liver disease [1]. In cases of obesity, adipose tissue macrophages change from the anti-inflammatory M2 phenotype to the pro-inflammatory M1 phenotype. These M1 macrophages produce inflammatory cytokines, such as TNF-α, IL-6, and IL-1β, that inhibit the ability of adipocytes to respond to insulin. M1 macrophages secrete cytokines, such as TNF-α and IL-6, to activate inflammatory pathways in insulin target cells, resulting in the activation of Jun N-terminal kinase (JNK) and inhibition of κB kinase β [1]. M2 adipose tissue macrophages, which are the major resident macrophages in lean adipose tissue, are characterized by high expression of IL-10 and CD206 [2]. The depletion of M1 macrophages normalizes sensitivity to insulin in obese mice, whereas the reduction of M2 macrophage numbers predisposes lean subjects to insulin resistance [2]. Indeed, studies show there is a close correlation between adipose tissue inflammation and metabolic diseases in obese humans [3,4]. Chronic inflammation is one of the major causes of vascular and kidney complications in patients with diabetes [5]. The inflammatory response could be stimulated by various mechanisms, including hyperglycemia-induced cell death, which increases the aggregation of macrophages in the kidney [6]. Blocking systemic or adipose tissue macrophage inflammation has been shown to improve insulin sensitivity in obese mice [7]. However, the regulatory mechanisms of M1/M2 polarization in adipose tissue under chronic inflammation and related complications in diabetes have not been well characterized.

Dipeptidyl peptidase 4 (DPP4) is a complex enzyme that is present on the surface of many cell types, including kidney, liver, pancreas, and fat cells, and also present as a soluble form in the circulation [8]. DPP4, a serine protease, cleaves the penultimate L-proline or L-alanine found at the N-terminal of several polypeptides, such as glucagon-like peptide 1 (GLP-1), glucose-dependent insulinotropic polypeptide (GIP), neuropeptides, and chemokines [9]. These peptides are rapidly degraded and inactivated by DPP4. In addition, DPP4 has been shown to exert a direct pro-inflammatory role in different cell types, including lymphocytes, macrophages, and smooth muscle cells [10,11]. Hematopoietic cells have been found to be a major source of circulating soluble DPP-4 (sDPP4). Moreover, activation of mouse and human T lymphocytes induced sDPP4 [12]. Previously, a study showed that plasma DPP4 levels and DPP4 mRNA expression in tissues increased progressively during the development of streptozotocin-induced type 1 diabetes [13]. Moreover, DPP4 expression in inflammatory cells such as dendritic cells and macrophages may have a significant role in modulating the adipose tissue inflammation in obesity through its nonenzymatic function [14]. Recently, a study reported that the circulating levels of endogenous soluble DPP4 were dissociated from the extent of systemic and white adipose tissue inflammation [15]. The study demonstrated that knockdown of adipocyte DPP4 expression decreased plasma DPP4 activity in older mice but did not change incretin levels or glucose homeostasis. In addition, the study found that the genetic disruption of hepatocyte DPP4 resulted in abrogation of obesity-associated increase of plasma DPP4 activity, repression of liver cytokine expression, and partly amelioration of inflammation in adipose tissue, however the incretin levels and glucose homeostasis were not affected [15]. These observations revealed complex biological processes in the regulation of soluble and enzymatic DPP4 as well as in the control of inflammation and glucose homeostasis.

In this study, we examined the involvement of adipose tissue M1/M2 cytokine expression in plasma DPP4 activity, systemic inflammation, and insulin resistance in a diabetic mouse model. We hypothesize that diabetes induces M1 but decreases M2 cytokine expression in adipose tissue. Increased M1 expression induces DPP4 activity in plasma and upregulates inflammatory mediators in liver that subsequently lead to systemic inflammation and insulin resistance in diabetes. Based on that, we first determined the changes of M1/M2 cytokine expression in adipose tissue of diabetic and non-diabetic mice. Furthermore, we explored whether diabetic plasma increases M1 expression and non-diabetic plasma decreases M1

expression in SVFs from adipose tissue of diabetes. Finally, we examined whether the injection of non-diabetic plasma reverses M1 cytokine expression, plasma DPP4 activity and CCL2 levels, systemic inflammation, and glucose intolerance in diabetes. Our findings provide a new insight that the inhibition of M1 cytokine expression in adipose tissue may represent a novel therapeutic approach for attenuating DPP4 activity, systemic inflammation, and insulin resistance in diabetic patients. Our findings provide a new insight that the inhibition of M1 cytokine expression in adipose tissue with non-diabetic plasma may represent a novel therapeutic approach for attenuating DPP4 activity, systemic inflammation, and insulin resistance in diabetic patients.

## Materials and methods

### Animals and treatments

$Lepr^{db/+}$ mice on the C57BL/6J genetic background were purchased from the Jackson Laboratory (Bar Harbor, ME). $Lepr^{db/+}$ mice were bred to obtain diabetic $Lepr^{db/db}$ (total n = 120) and non-diabetic $Lepr^{+/+}$ mice (total n = 80). $Lepr^{db/db}$ mice that have a mutation in the gene encoding the leptin receptor become obese at 3 to 4 weeks of age, and elevated plasma insulin and blood sugar can be observed in $Lepr^{db/db}$ mice at four to eight weeks of age. $Jnk1^{-/-}$ (c-Jun N-terminal kinases 1 knockout) mice on the C57BL/6J background were transferred from Dr. Karin's laboratory (University of California, San Diego, CA, USA). $Jnk1^{-/-}$ mice were crossbred with $Lepr^{db/+}$ mice to obtain $Lepr^{db/+}$- $Jnk1^{-/-}$ mice, and $Lepr^{db/+}$-$Jnk1^{-/-}$ mice were then bred to obtain $Lepr^{db/db}$-$Jnk1^{-/-}$ mice (total n = 30). All mice had ad libitum access to water and food and were fed a standard laboratory diet (1324 TPF; Atromin; Large Germany; 11.9 kJ/g, 19% crude protein, 4% crude fat, 6% crude fiber).

### Ethics statement

The Institutional Animal Care and Use Committee (IACUC) of Kaohsiung Veterans General Hospital has approved this study (Approval Number: 2020-A042), and animal experiments were performed in agreement with Animal Experimentation Regulations of Kaohsiung Veterans General Hospital. Animals were observed every 12 hours. All efforts were made to reduce suffering of animals. Personnel conducting mice treatment received training and competency testing from Kaohsiung Veterans General Hospital. Mice were euthanized with carbon dioxide when they were found in a moribund state as identified by inability to maintain upright with or without labored breathing and cyanosis, >15% loss of body weight, inability to obtain food or water, or unresponsive to normal physical manipulation. Once any animal reached endpoint criteria, the amount of time elapsed before euthanasia was less than 12 hours. No mortality occurred outside of planned euthanasia or humane endpoints.

### Preparation of SVFs

SVFs were prepared from $Lepr^{db/db}$ or $Lepr^{+/+}$ mice. Vascular adipose tissue was isolated from bilateral inguinal adipose tissue of diabetic and non-diabetic mice and minced into small pieces. These tissues were then digested with collagenase 8 (Sigma-Aldrich, Cat# C2139) in ice-cold HBSS (2 mg/ml) for 15 minutes at 37°C. After passing through a 100 μm cell strainer, cells were centrifuged at 1,200 rpm for 10 min, and the cell pellets were retrieved as SVFs.

### *In vitro* treatment of SVFs

For *in vitro* treatment, 1ml of 10% plasma was added to the equal amount of SVFs ($1 \times 10^9$ cells/ml), and the mixture was incubated at 37°C for 4 hr as previously suggested [11]. Samples

were then centrifuged at 2,500 rpm for 5 min and washed with phosphate buffered saline (PBS). After centrifugation, the pellets were harvested and subjected to analysis.

### *In vivo* injection of plasma into inguinal adipose tissue

Male *Lepr*$^{db/db}$ mice were randomly divided into the following three groups: Group I received 1ml PBS injection into adipose tissue over the bilateral inguinal area; Group II received an injection of 1ml 10% plasma from *Lepr*$^{db/db}$ mice into adipose tissue over the bilateral inguinal area; and Group III received an injection of 1ml 10% plasma from *Lepr*$^{+/+}$ mice into adipose tissue over the bilateral inguinal area. Seven days after injection, the animals were euthanized and liver, adipose tissue, and plasma were harvested for analysis.

### RNA isolation and quantitative real-time polymerase chain reaction (Q-PCR)

Total RNA was purified from mouse samples using total RNA Miniprep Purification Kits (GeneMark) according to the manufacturer's instructions. Total RNA was then reverse-transcribed into cDNA using a RT kit (Invitrogen, Carlsbad, CA, Lot# 2234812). Primer pairs were synthesized by Integrated DNA Technologies (Coralville, IA). Primer sequences are listed in S1 Table. For Q-PCR assay, 200 ng of the cDNA template was added to 25 μl of mixture containing 12.5 μl of 2× Fast SYBR Green Master Mix (Applied biosystems, Cat# 4385612), 2.5 μl of sense and anti-sense primers (25 μM), 2 μl of sample, and 8 μl of sterile water. The amplification was performed in a StepOnePlus™ Real-Time PCR System (Applied Biosystems 7300).

### Western immunoblots

The protein expression of DPP4 (GeneTex, Cat# GTX84602), phospho-Akt (Cell signaling, Cat# 4060), Akt (Cell signaling, Cat# 4691), phospho-p38 (Cell signaling, Cat# 9211), p38 (Cell signaling, Cat# 9212), FMO3 (Novus, Cat# NBP1-33583), ICAM-1 (Abnova, Cat# E5081) in the liver as well as JNK (Cell signaling, Cat# 9252) and pJNK (Cell signaling, Cat# 9251) in the SVFs was detected by antibodies. The harvested tissues were weighed and homogenized in protein extraction buffer (Sigma), containing proteinase inhibitor cocktail (Roche). The homogenized samples were subjected to SDS-PAGE at 50 to 100V for 2 hours, and the proteins were then transferred onto the nitrocellulose membrane. The membranes were blocked with 5% non-fat milk in TBST buffer (10 mM Tris-HCl, pH 7.5, 150 mM NaCl and 1.2% Tween 20) for 1 hour and incubated with specific primary antibodies at room temperature for 1 hour. The membranes were then washed with TBST buffer followed by incubation with the secondary antibodies. After washing with TBST buffer, the protein bands were identified by enhanced chemiluminescence (ECL) detection reagent (Millipore).

### Kupffer cell purification

The liver was perfused *in situ* through the portal vein with Ca$^{2+}$- and Mg$^{2+}$-free phosphate-buffered saline containing 10 mM ethylenediaminetetraacetic acid at 37°C for 5 minutes. Subsequently, perfusion was performed with HBSS containing 0.1% collagenase IV (Sigma) at 37°C for 5 minutes. After digestion, the liver was excised and the suspension was filtered. The sample was centrifuged twice at 50g at 4°C for 1 minute. The supernatant was harvested and centrifuged at 300g for 5 minutes, and the pellet was collected and resuspended with buffer. The cell suspension was then layered on top of a density cushion of 30/60 discontinuous Percoll (Pharmacia) and centrifuged at 900g for 15 minutes to obtain the Kupffer cell fraction [16].

### Plasma DPP4 activity *n*

Cardiac blood (500 μl) was collected and centrifuged to harvest plasma. Harvested plasma was stored at -20˚C until assayed. Plasma DPP4 activities were detected by DPP4 assay kit (BioVision, Milpitas, CA, Cat# K779-100). DPP4 cleaves substrates to release the quenched fluorescent group, AMC (7-Amino-4-Methyl Coumarin), which was detected at Ex/Em = 360/460 nm using a fluorescence reader.

### Enzyme-linked immunosorbent assay (ELISA)

The mouse ELISA kit (R&D, Minneapolis, MN, Cat# DY479-05) was used for CCL2 assay. The blood was centrifuged at 1000×*g*, 4˚C for 15 minutes and the serum was then collected for use. The ELISA plates were coated with 100 μl capture antibody at 4˚C overnight. After wash, 200 μl of assay dilution buffer were added for blocking at room temperature for 1 hour. The samples and serial dilutions of standards were added and incubated at 4˚C for overnight. After incubating with the detection antibody, avidin-HRP was added and incubated at room temperature for 30 minutes. The substrate, 3,3′,5,5′-tetramethylbenzidine (TMB), was then added and incubated for 15 minutes. 100 μl of stop solution were added to stop reaction, and the plate was then subjected to measurement of absorbance at 450 nm by using an ELISA reader.

### Intraperitoneal glucose tolerance tests (IPGTTs)

Glucose tolerance was assessed using intraperitoneal glucose tolerance tests (IPGTTs). Mice were tested in the morning after a 15-hour fasting period. Tail blood glucose was measured before and at 5, 15, 30, 45, 60, 75, 90, and 120 minutes after intraperitoneal injection of glucose (2 g/kg body weight; Sigma). Plasma glucose was measured using a glucose meter (Accu-check performa; Roche, Switzerland).

### Statistical analysis

Data were analyzed by unpaired *t* test for the comparisons between two groups or by one-way analysis of variance (ANOVA) followed by Tukey's Multiple comparison test for the comparisons between multiple groups. All values in the figures and texts were expressed as mean ± standard error of the mean, and *p* values less than 0.05 are considered statistically significant.

## Results

### SVFs from *Lepr$^{db/db}$* mice exhibit increased M1 expression and SVFs from *Lepr$^{+/+}$* mice exhibit increased M2 expression

To study M1/M2 polarization in the adipose tissue of diabetes and non-diabetes, we examined M1/M2 cytokine mRNA expression in SVFs of inguinal adipose tissue from type 2 diabetic (*Lepr$^{db/db}$*) mice and control non-diabetic (*Lepr$^{+/+}$*) mice. SVFs from *Lepr$^{db/db}$* mice demonstrated a significant induction of TNF-α, IL-6, IL-1β, CCL2, and DPP4 mRNA expression compared with those from *Lepr$^{+/+}$* mice. In contrast, SVFs from *Lepr$^{db/db}$* mice exhibited a significant reduction of IL-10 compared with those from *Lepr$^{+/+}$* mice (Fig 1). Collectively, these results indicate that diabetes induces M1 but reduces M2 cytokine expression in the adipose tissue.

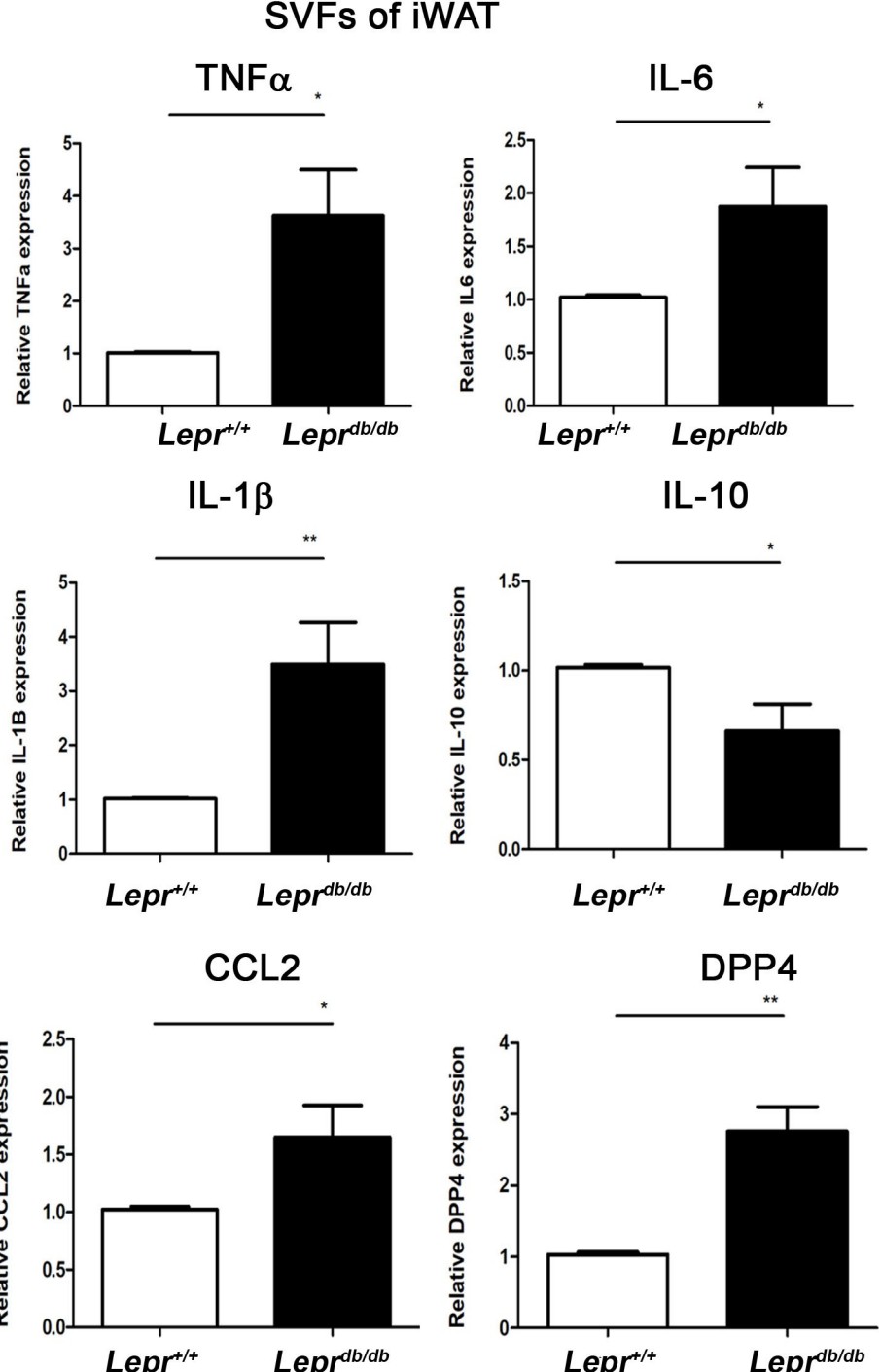

**Fig 1. SVFs of adipose tissue from *Lepr^db/db* mice demonstrate an increased M1 but decreased M2 signature compared with those from *Lepr^+/+* mice.** Inguinal white adipose tissue was harvested from *Lepr^db/db* and *Lepr^+/+* mice. SVFs were then purified from adipose tissue and subjected to Q-PCR analysis for mRNA expression of TNF-α, IL-6, IL-1β, IL-10, CCL2, and DPP4. N = 5/group. $*p<0.05$, $**p<0.01$.

## Diabetic plasma induces M1 expression and non-diabetic plasma induces M2 expression in SVFs of *Lepr$^{db/db}$* mice

To examine the effects of diabetic and non-diabetic plasma on M1/M2 polarization in the adipose tissue of diabetes, SVFs were harvested from the adipose tissue of *Lepr$^{db/db}$* mice and treated with plasma from *Lepr$^{db/db}$* or *Lepr$^{+/+}$* mice *in vitro*. Treatment of *Lepr$^{db/db}$* SVFs with *Lepr$^{db/db}$* plasma demonstrated a significant upregulation of TNF-α, IL-6, IL-1β, CCL2, and DPP4 mRNA expression compared with those treated with PBS (Fig 2). On the contrary, treatment of *Lepr$^{db/db}$* SVFs with *Lepr$^{+/+}$* plasma resulted in a significant downregulation of aforementioned inflammatory mediators and upregulation of IL-10 mRNA expression compared with those treated with *Lepr$^{db/db}$* plasma (Fig 2). Altogether, these results demonstrate that diabetic plasma induces M1 expression, however non-diabetic plasma induces M2 expression in the adipose tissue of diabetic mice.

## Injection of plasma from *Lepr$^{+/+}$* mice into adipose tissue of *Lepr$^{db/db}$* mice reduces M1 expression *in vivo*

To further evaluate the effects of diabetic and non-diabetic plasma on M1/M2 polarization of adipose tissue in diabetic mice, plasma harvested from *Lepr$^{db/db}$* and *Lepr$^{+/+}$* mice was injected into the adipose tissue near bilateral inguinal area in *Lepr$^{db/db}$* mice, and SVFs were then isolated from the adipose tissue followed by Q-PCR analysis to determine cytokine expression. Our results showed that *Lepr$^{db/db}$* mice injected with PBS exhibited increased expression of TNF-α, IL-6, IL-1β, CCL2, and DPP4 mRNA expression in SVFs compared to *Lepr$^{+/+}$* mice injected with PBS (Fig 3A). Notably, injection of *Lepr$^{+/+}$* plasma significantly induced IL-10 but decreased TNF-α, IL-6, IL-1β, and DPP4 mRNA expression in SVFs of *Lepr$^{db/db}$* mice compared with those injected with *Lepr$^{db/db}$* plasma (Fig 3A). Taken altogether, these results demonstrate that non-diabetic plasma reduces M1 cytokine but increases M2 cytokine expression in diabetic adipose tissue.

## Injection of plasma from *Lepr$^{+/+}$* mice into adipose tissue of *Lepr$^{db/db}$* mice decreases JNK activation and DPP4 protein expression in adipose tissue of *Lepr$^{db/db}$* mice

To investigate the effects of diabetic and non-diabetic plasma on JNK activation and DPP4 expression in the adipose tissue of diabetes, plasma harvested from *Lepr$^{db/db}$* and *Lepr$^{+/+}$* mice was injected into the adipose tissue of *Lepr$^{db/db}$* mice, and JNK activation and DPP4 protein expression in SVFs of adipose tissue were determined. SVFs from *Lepr$^{db/db}$* mice displayed increased JNK activation and DPP4 protein expression compared with those from *Lepr$^{+/+}$* mice (Fig 3B). In contrast, injection of *Lepr$^{+/+}$* plasma mice into the adipose tissue of *Lepr$^{db/db}$* mice significantly decreased JNK activation and DPP4 protein expression in SVFs of *Lepr$^{db/db}$* mice compared with those receiving *Lepr$^{db/db}$* plasma. Altogether, these results suggest that diabetes induces JNK activation and DPP4 protein expression and non-diabetic plasma diminishes diabetes-induced JNK activation and DPP4 protein expression in the adipose tissue of diabetes.

## Injection of plasma from *Lepr$^{+/+}$* mice into adipose tissue of *Lepr$^{db/db}$* mice decreases inflammatory cytokine expression in the liver of *Lepr$^{db/db}$* mice

To further explore the effects of diabetic and non-diabetic plasma on cytokine expression in the liver of diabetes, plasma harvested from *Lepr$^{db/db}$* and *Lepr$^{+/+}$* mice was injected into the adipose tissue of *Lepr$^{db/db}$* mice, and the cytokine expression in the liver of *Lepr$^{db/db}$* mice was

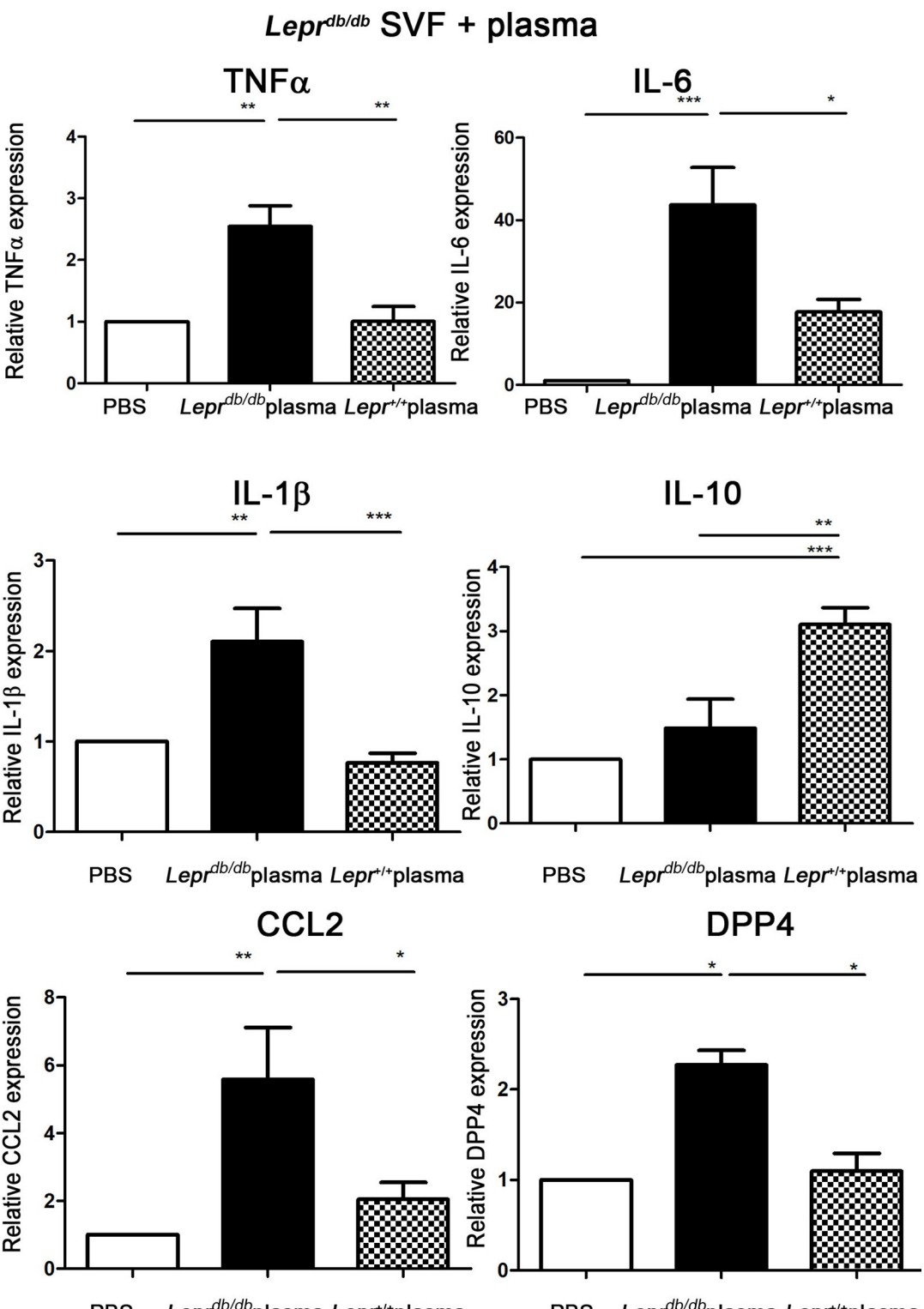

**Fig 2. Diabetic plasma induces M1 expression and non-diabetic plasma induces M2 expression in SVFs from *Lepr*^db/db^ mice.** Plasma harvested from *Lepr*^db/db^ or *Lepr*^+/+^ mice were diluted with PBS to 10% and added to the equal amount of SVFs isolated from adipose tissue of *Lepr*^db/db^ mice followed by *in vitro* culture at 37˚C. After 4 hours of incubation, cells were then collected and subjected to Q-PCR analysis for mRNA expression of TNF-α, IL-6, IL-1β, IL-10, CCL2, and DPP4. N = 5/group. $^*p < 0.05$, $^{**}p < 0.01$, $^{***}p < 0.001$.

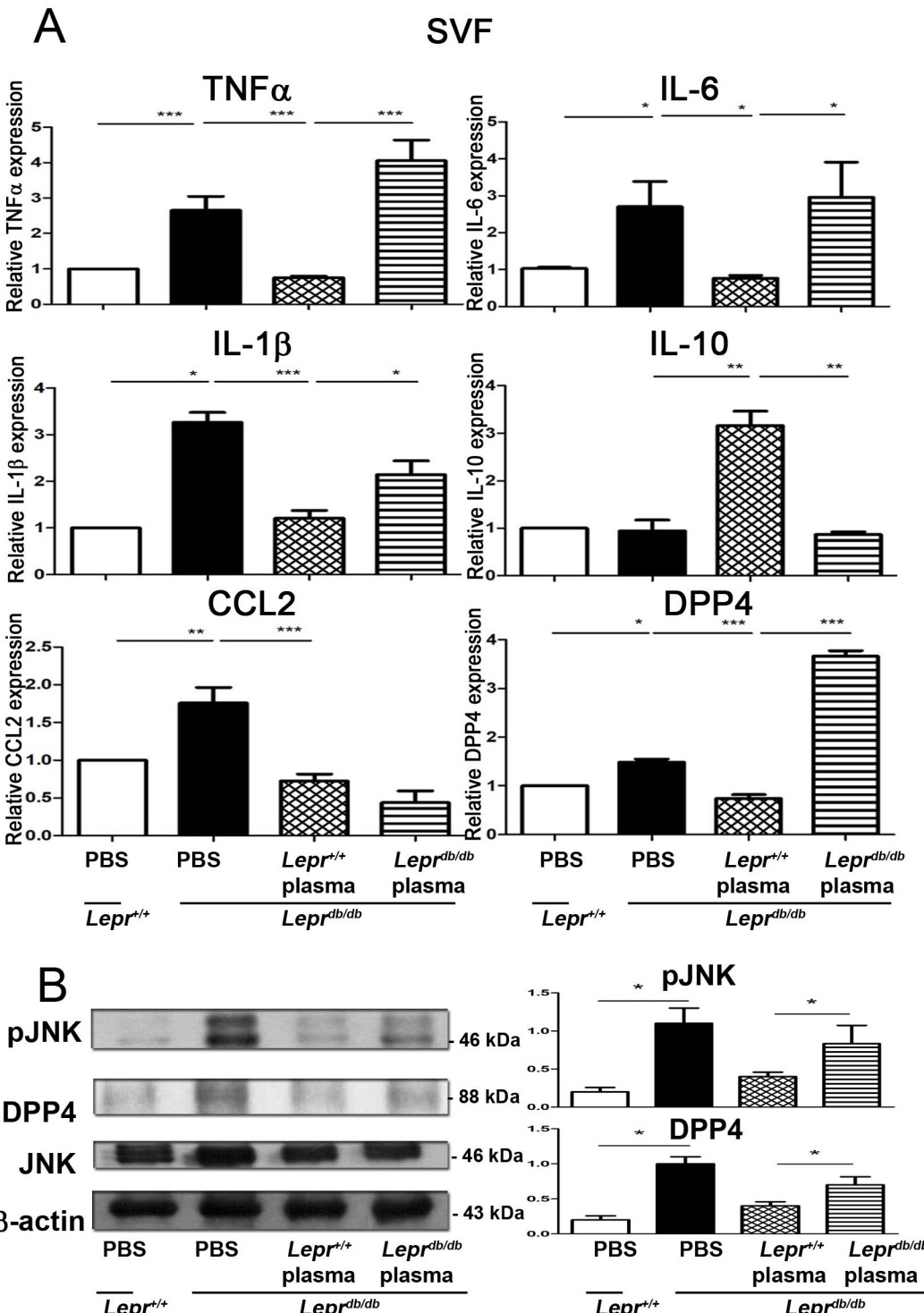

**Fig 3. Injection of plasma from *Lepr*$^{+/+}$ mice into adipose tissue of *Lepr*$^{db/db}$ mice reduces M1 mRNA expression as well as pJNK and DPP4 protein expression in adipose tissue of *Lepr*$^{db/db}$ mice *in vivo*.** Plasma was harvested from *Lepr*$^{db/db}$ or *Lepr*$^{+/+}$ mice and diluted with PBS to 10%. One ml of 10% plasma was injected into adipose tissue over bilateral inguinal area of *Lepr*$^{db/db}$ mice. *Lepr*$^{db/db}$ and *Lepr*$^{+/+}$ mice injected with PBS served as controls. Seven days after injection, animals were euthanized and adipose tissue was harvested. (A) SVFs were then isolated from adipose tissue and subjected to Q-PCR analysis to measure mRNA expression of TNFα, IL-6, IL-1β, IL-10, CCL2, DPP4. (B) SVFs were also subjected to western blots analysis to detect protein expression of phospho-JNK, total JNK, and DPP4. The expression level of phospho-JNK and DPP4 was also quantified. N = 6/group. $^{*}p<0.05$; $^{**}p<0.01$; $^{***}p<0.001$.

evaluated. *Lepr*$^{db/db}$ mice exhibited increased mRNA expression of ICAM, FMO3, IL-1β, iNOS, TNF-α, IL-6, and DPP4 in the liver compared with *Lepr*$^{+/+}$ mice (Fig 4A). Notably, injection of *Lepr*$^{+/+}$ plasma into the adipose tissue of *Lepr*$^{db/db}$ mice largely decreased mRNA expression of ICAM, FMO3, IL-1β, iNOS, TNF-α, IL-6, and DPP4 in the liver of *Lepr*$^{db/db}$ mice compared with those receiving the injection of *Lepr*$^{db/db}$ plasma (Fig 4A). Altogether, these results demonstrate that non-diabetic plasma suppresses the expression of inflammatory mediators in the liver of diabetic mice.

## Injection of plasma from *Lepr*$^{+/+}$ mice decreases ICAM, FMO3, and DPP4 protein expression in the liver of *Lepr*$^{db/db}$ mice

To elucidate the effects of diabetic and non-diabetic plasma on ICAM, FMO3, and DPP4 protein expression in the liver of diabetes, plasma from *Lepr*$^{db/db}$ and Lepr$^{+/+}$ mice was injected into the adipose tissue of *Lepr*$^{db/db}$ mice, and the protein expression of ICAM, DPP4, and FMO3 in the liver was examined. *Lepr*$^{db/db}$ mice displayed increased liver ICAM, FMO3, and DPP4 protein expression compared with *Lepr*$^{+/+}$ mice (Fig 4B). Injection of *Lepr*$^{+/+}$ plasma into the adipose tissue significantly decreased ICAM, FMO3, and DPP4 protein expression in the liver of *Lepr*$^{db/db}$ mice compared with those receiving *Lepr*$^{db/db}$ plasma. Collectively, these results indicate that diabetes induces ICAM, FMO3, and DPP4 expression in the liver, and non-diabetic plasma injected into the adipose tissue reduces the expression of those proteins in the liver of diabetic mice.

## Injection of plasma from *Lepr*$^{+/+}$ mice to adipose tissue of *Lepr*$^{db/db}$ mice decreases TNF-α, IL-6, INOS, and DPP4 mRNA expression of Kupffer cells in *Lepr*$^{db/db}$ mice

To further examine the effects of diabetic and non-diabetic plasma on inflammatory cytokine expression in Kupffer cells of diabetes, plasma harvested from *Lepr*$^{db/db}$ and *Lepr*$^{+/+}$ mice was injected into the adipose tissue of *Lepr*$^{db/db}$ mice, and Kupffer cells were then isolated from the liver of *Lepr*$^{db/db}$ mice and subjected to Q-PCR analysis to determine mRNA expression of inflammatory cytokines. Kupffer cells isolated from *Lepr*$^{db/db}$ mice displayed increased mRNA expression of TNF-α, INOS, DPP4, and IL-6 compared with those isolated from *Lepr*$^{+/+}$ mice (Fig 5). In contrast, Kupffer cells isolated from *Lepr*$^{db/db}$ mice treated with *Lepr*$^{+/+}$ plasma showed repressed mRNA expression of aforementioned inflammatory mediators compared with those isolated from *Lepr*$^{db/db}$ mice treated with *Lepr*$^{db/db}$ plasma (Fig 5). Altogether, these results demonstrate that injection of non-diabetic plasma into the adipose tissue is capable of suppressing inflammatory mediator production in Kupffer cells of diabetic mice.

## *Lepr*$^{db/db}$ mice exhibit increased plasma DPP4 activity, whereas injection of *Lepr*$^{+/+}$ plasma decreases plasma DPP4 levels in *Lepr*$^{db/db}$ mice

To examine whether there is a difference in plasma DPP4 activity between diabetic and non-diabetic mice, we measured plasma DPP4 activity in *Lepr*$^{db/db}$ and *Lepr*$^{+/+}$ mice of both sexes. We observed that *Lepr*$^{db/db}$ mice exhibited higher plasma DPP4 activity compared with *Lepr*$^{+/+}$ mice in both males and females (Fig 6A). We then further determine the effects of diabetic and non-diabetic plasma on plasma DPP4 activity in diabetes. Plasma harvested from *Lepr*$^{db/db}$ or *Lepr*$^{+/+}$ mice was injected into the adipose tissue of *Lepr*$^{db/db}$ mice, and plasma DPP4 activity was then assessed. Our results showed that injection of *Lepr*$^{+/+}$ plasma into the adipose tissue of *Lepr*$^{db/db}$ mice resulted in a significant reduction of plasma DPP4 activity in *Lepr*$^{db/db}$ mice compared with those receiving *Lepr*$^{db/db}$ plasma ($145.5 \pm 11.61$ vs. $199 \pm 7.1$ pmol/min/ml×10$^{-3}$)

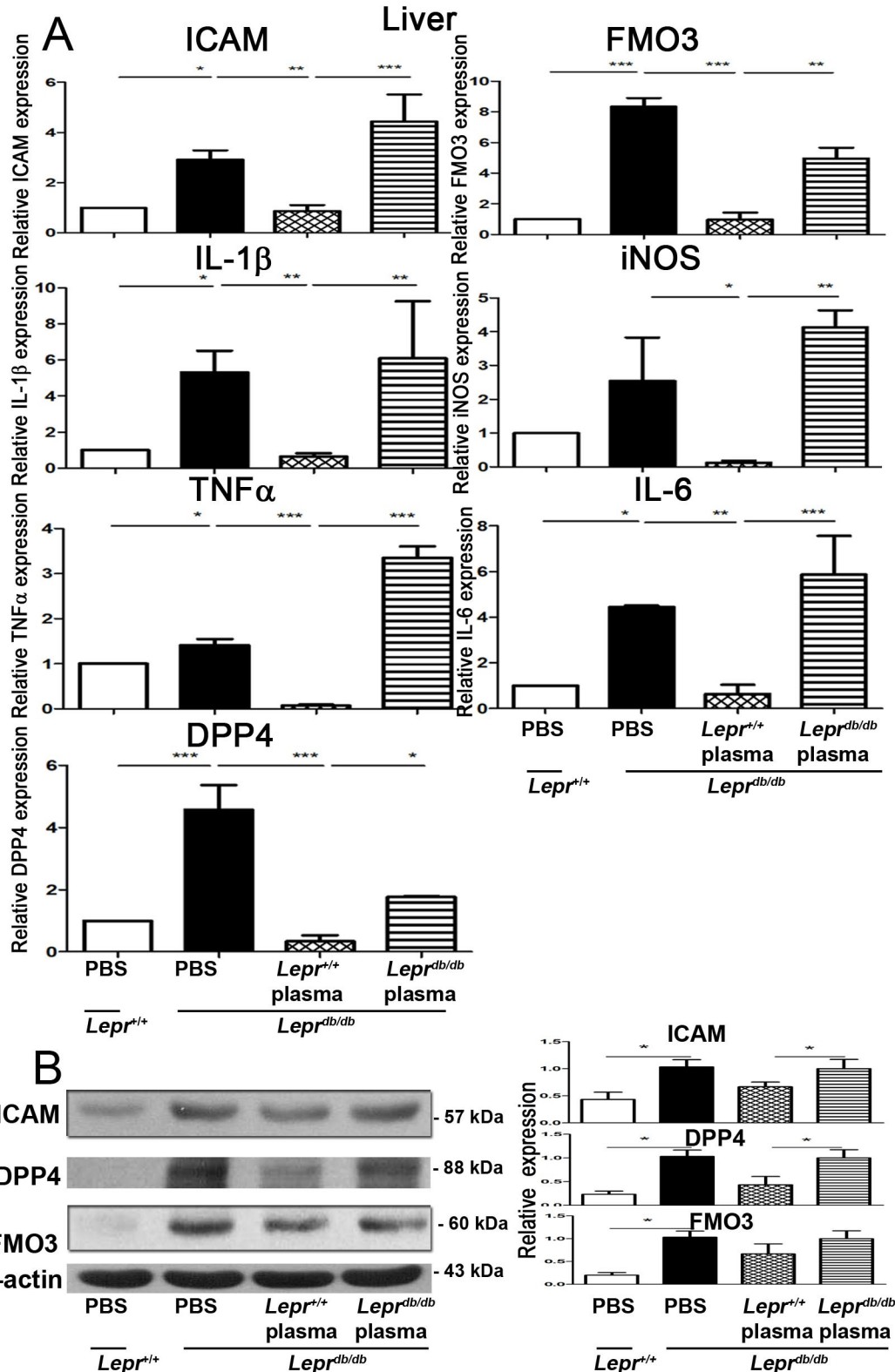

**Fig 4. Injection of plasma from *Lepr*[+/+] mice into adipose tissue of *Lepr*[db/db] mice decreases inflammatory cytokine expression as well as ICAM, FMO3, and DPP4 protein expression in the liver of *Lepr*[db/db] mice.** Plasma harvested from *Lepr*[db/db] or *Lepr*[+/+] mice was diluted with PBS to 10%. One ml of 10% plasma was then injected to adipose tissue of *Lepr*[db/db] mice. *Lepr*[db/db] and *Lepr*[+/+] mice were injected with PBS to serve as controls. Seven days after injection, animals were

euthanized, and livers were then harvested and subjected to (A) Q-PCR analysis for ICAM, FMO3, IL-1β, iNOS, TNFα, IL-6, and DPP4 mRNA expression or (B) western blot analysis for ICAM, DPP4, and FMO3 protein expression. The expression level of ICAM, DPP4, and FMO3 was also quantified. N = 6/group. *$p<0.05$; **$p<0.01$; ***$p<0.001$.

(Fig 6B). Collectively, these results demonstrate that $Lepr^{db/db}$ mice exhibit increased plasma DPP4 activity, and non-diabetic plasma is capable of repressing plasma DPP4 activity in $Lepr^{db/db}$ mice.

## Injection of plasma from $Lepr^{+/+}$ mice into adipose tissue of $Lepr^{db/db}$ mice decreases plasma CCL2 levels in $Lepr^{db/db}$ mice

To identify the effects of diabetic and non-diabetic plasma on plasma CCL2 levels in diabetes, we harvested plasma from $Lepr^{db/db}$ and $Lepr^{+/+}$ mice, and injected them into adipose tissue of $Lepr^{db/db}$ mice. Plasma CCL2 levels were then measured. Our results showed that plasma CCL2 levels were significantly higher in $Lepr^{db/db}$ mice compared to $Lepr^{+/+}$ mice ($53.05 \pm 1.74$ vs. $7.4 \pm 1.26$ pg/ml) (Fig 6C). On the contrary, injection of $Lepr^{+/+}$ plasma into the adipose tissue of $Lepr^{db/db}$ mice significantly decreased plasma CCL2 levels in $Lepr^{db/db}$ mice when compared to those injected with $Lepr^{db/db}$ plasma ($16.07 \pm 2.36$ vs. $39.47 \pm 6.6$ pg/ml) (Fig 6C).

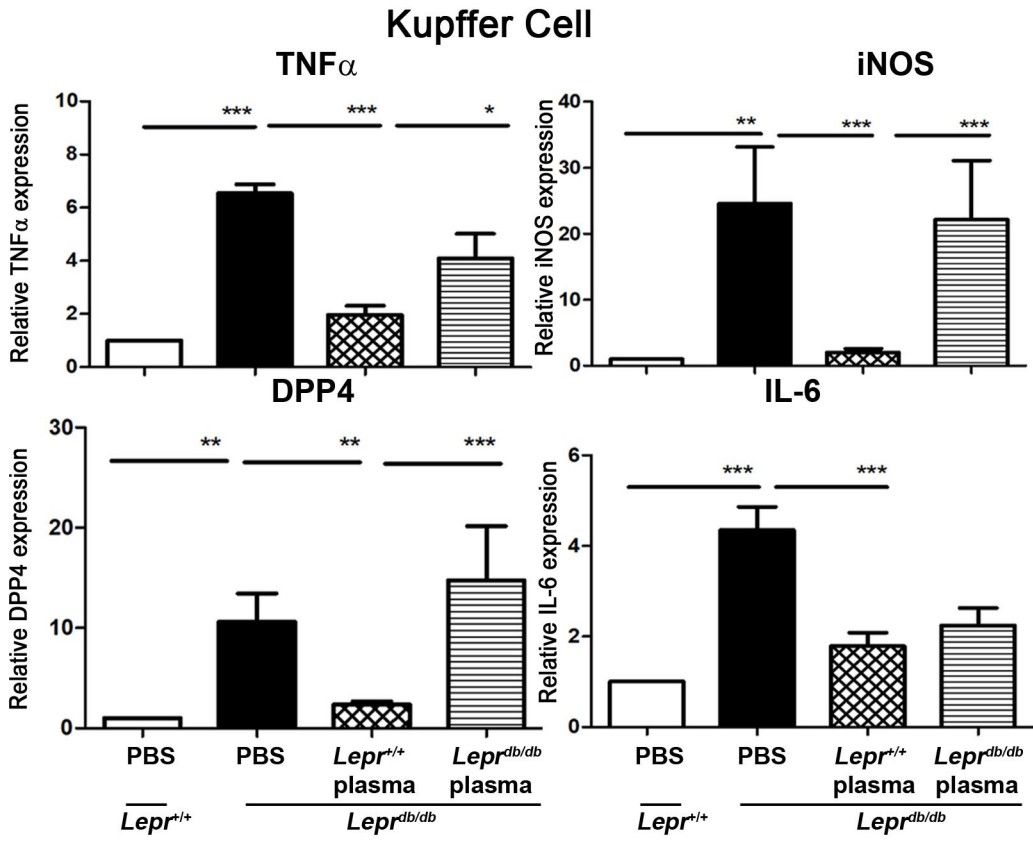

**Fig 5. Injection of plasma from $Lepr^{+/+}$ mice into adipose tissue of $Lepr^{db/db}$ mice decreases TNF-α, IL-6, iNOS, and DPP4 mRNA expression in Kupffer cells of $Lepr^{db/db}$ mice.** Plasma harvested from $Lepr^{db/db}$ or $Lepr^{+/+}$ mice was diluted with PBS to 10%, and one ml of 10% plasma was injected into adipose tissue of $Lepr^{db/db}$ mice. $Lepr^{db/db}$ and $Lepr^{+/+}$ mice were also injected with PBS to serve as controls. Seven days after injection, animals were euthanized and livers were harvested. Kupffer cells were then isolated from livers and subjected to Q-PCR analysis for mRNA expression of TNFα, iNOS, DPP4, and IL-6. N = 6/group. *$p<0.05$; **$p<0.01$; ***$p<0.001$.

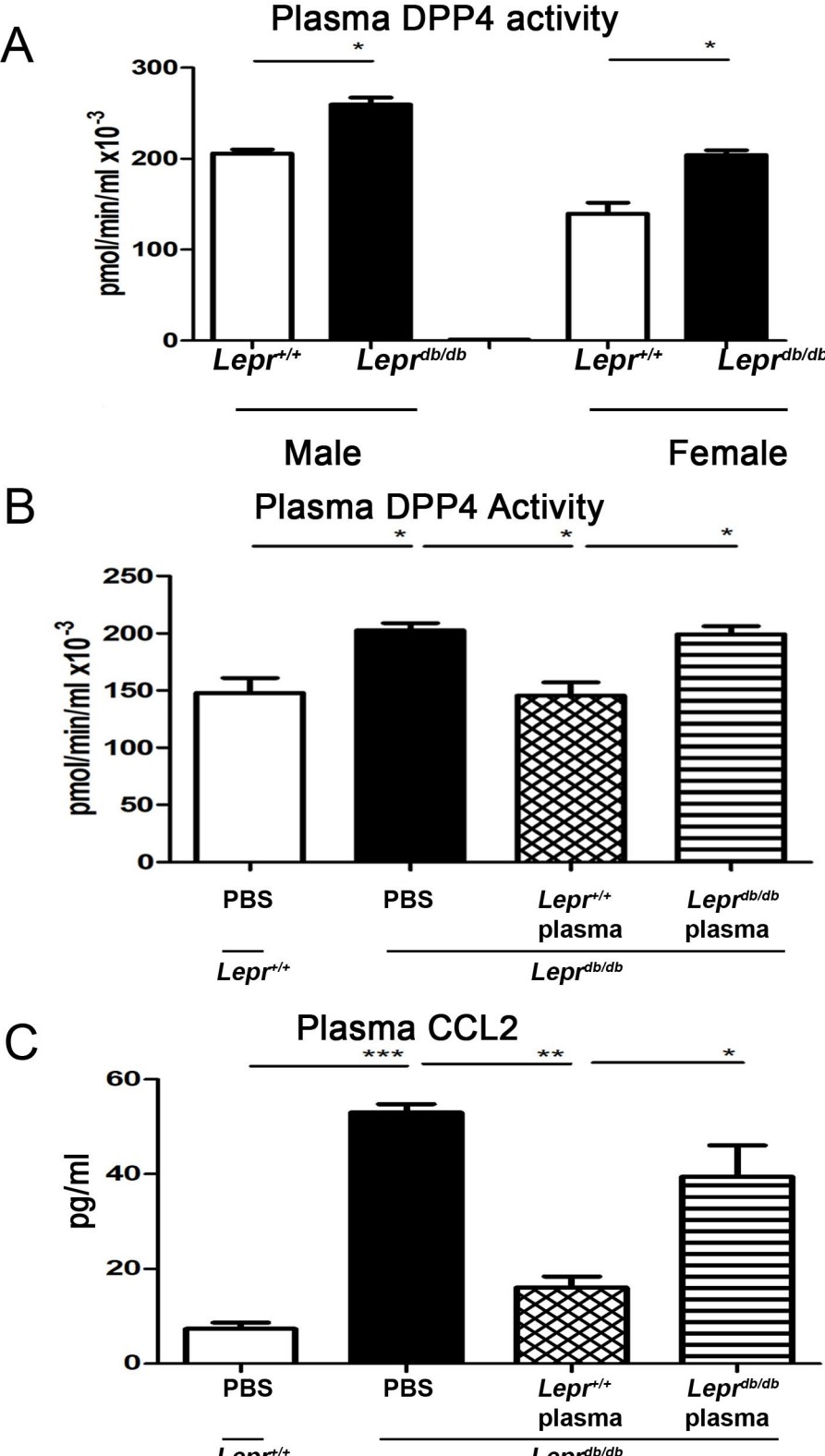

**Fig 6. Diabetic *Lepr^db/db* mice demonstrate increased plasma DPP4 activity and injection of *Lepr^+/+* plasma decreases plasma DPP4 activity and CCL2 levels in *Lepr^db/db* mice.** (A) Cardiac blood was collected from male and female *Lepr^db/db* and *Lepr^+/+* mice followed by plasma extraction, and plasma DPP4 activity was then detected by DPP4

assay kit. (B and C) *Lepr*$^{db/db}$ mice were injected with 1ml of 10% plasma harvested from *Lepr*$^{db/db}$ or *Lepr*$^{+/+}$ mice. *Lepr*$^{db/db}$ and *Lepr*$^{+/+}$ mice were also injected with PBS to serve as controls. After seven days, animals were euthanized and plasma was harvested to detect DPP4 activity and CCL2 level. N = 5/group. $^{*}p<0.05$; $^{**}p<0.01$; $^{***}p<0.001$.

Altogether, these results suggest that diabetes increases plasma CCL2 levels, and non-diabetic plasma possesses a suppressive effect on plasma CCL2 levels in diabetes.

## Injection of plasma from *Lepr*$^{+/+}$ mice increases Akt activation in *Lepr*$^{db/db}$ mice following insulin administration

To investigate whether non-diabetic plasma affects insulin resistance in diabetes, *Lepr*$^{db/db}$ mice were subjected to *Lepr*$^{+/+}$ plasma injection, and *Lepr*$^{+/+}$, *Lepr*$^{db/db}$, and *Lepr*$^{+/+}$ plasma-injected *Lepr*$^{db/db}$ mice were administered with or without insulin to determine the level of Akt phosphorylation in the liver. Akt phosphorylation was only observed in the liver of *Lepr*$^{+/+}$ mice but not *Lepr*$^{db/db}$ mice following insulin administration, indicating that *Lepr*$^{db/db}$ mice did not respond to insulin treatment (Fig 7A and 7B). However, Akt phosphorylation was able to be observed in the liver of *Lepr*$^{db/db}$ mice injected with *Lepr*$^{+/+}$ plasma following insulin administration (Fig 7A and 7B), suggesting that *Lepr*$^{+/+}$ plasma promotes insulin response in *Lepr*$^{db/db}$ mice. In addition, *Lepr*$^{db/db}$ mice exhibited increased DPP4 expression in the liver compared to *Lepr*$^{+/+}$ mice, and the injection of *Lepr*$^{+/+}$ plasma was able to suppress DPP4 expression in the liver of *Lepr*$^{db/db}$ mice. Collectively, these results indicate that diabetes induces DPP4 expression and suppress insulin-induced Akt activation in the liver, and the injection of non-diabetic plasma into the adipose tissue of diabetic animals reverses those effects.

## Injection of plasma from *Lepr*$^{+/+}$ mice reverses glucose intolerance in *Lepr*$^{db/db}$ mice

To further examine the effect of injection of *Lepr*$^{+/+}$ plasma on diabetes-induced glucose intolerance, we performed glucose tolerance test in *Lepr*$^{+/+}$, *Lepr*$^{db/db}$, and *Lepr*$^{+/+}$ plasma-injected *Lepr*$^{db/db}$ mice. After fasting, *Lepr*$^{+/+}$ mice showed an increased blood glucose level after glucose administration, however, the blood glucose level returned to the normal level at 2 hours post administration. In contrast, *Lepr*$^{db/db}$ mice maintained a high blood glucose level even at 2 hours after glucose administration. Notably, *Lepr*$^{+/+}$ plasma-treated *Lepr*$^{db/db}$ mice, unlike *Lepr*$^{db/db}$ mice, exhibited a reduced blood glucose level post glucose administration (Fig 7C). Altogether, these results demonstrate that diabetes induces glucose intolerance, however non-diabetic plasma ameliorates impaired glucose tolerance in diabetic mice.

## Diabetic plasma induces M1 expression in adipose tissue from *Lepr*$^{db/db}$ mice but not in adipose tissue from *Lepr*$^{db/db}$-*Jnk1*$^{-/-}$ mice

To examine whether JNK activation plays an essential role in adipose tissue M1 expression in diabetes, SVFs harvested from adipose tissue of *Lepr*$^{db/db}$ and *Lepr*$^{db/db}$-*Jnk1*$^{-/-}$ mice were treated with PBS or *Lepr*$^{db/db}$ plasma *in vitro*. *Lepr*$^{db/db}$ SVFs treated with *Lepr*$^{db/db}$ plasma demonstrated increased mRNA expression of TNF-α, IL-6, IL-1β, CCL2, and DPP4 compared with *Lepr*$^{db/db}$ SVFs treated with PBS (Fig 8). In contrast, *Lepr*$^{db/db}$-*Jnk1*$^{-/-}$ SVFs treated with *Lepr*$^{db/db}$ plasma did not exhibit increased mRNA expression of aforementioned inflammatory mediators compared with *Lepr*$^{db/db}$-*Jnk1*$^{-/-}$ SVFs treated with PBS (Fig 8). Altogether, these results suggest that JNK activation may play an essential role in promoting adipose tissue M1 polarization in diabetic mice.

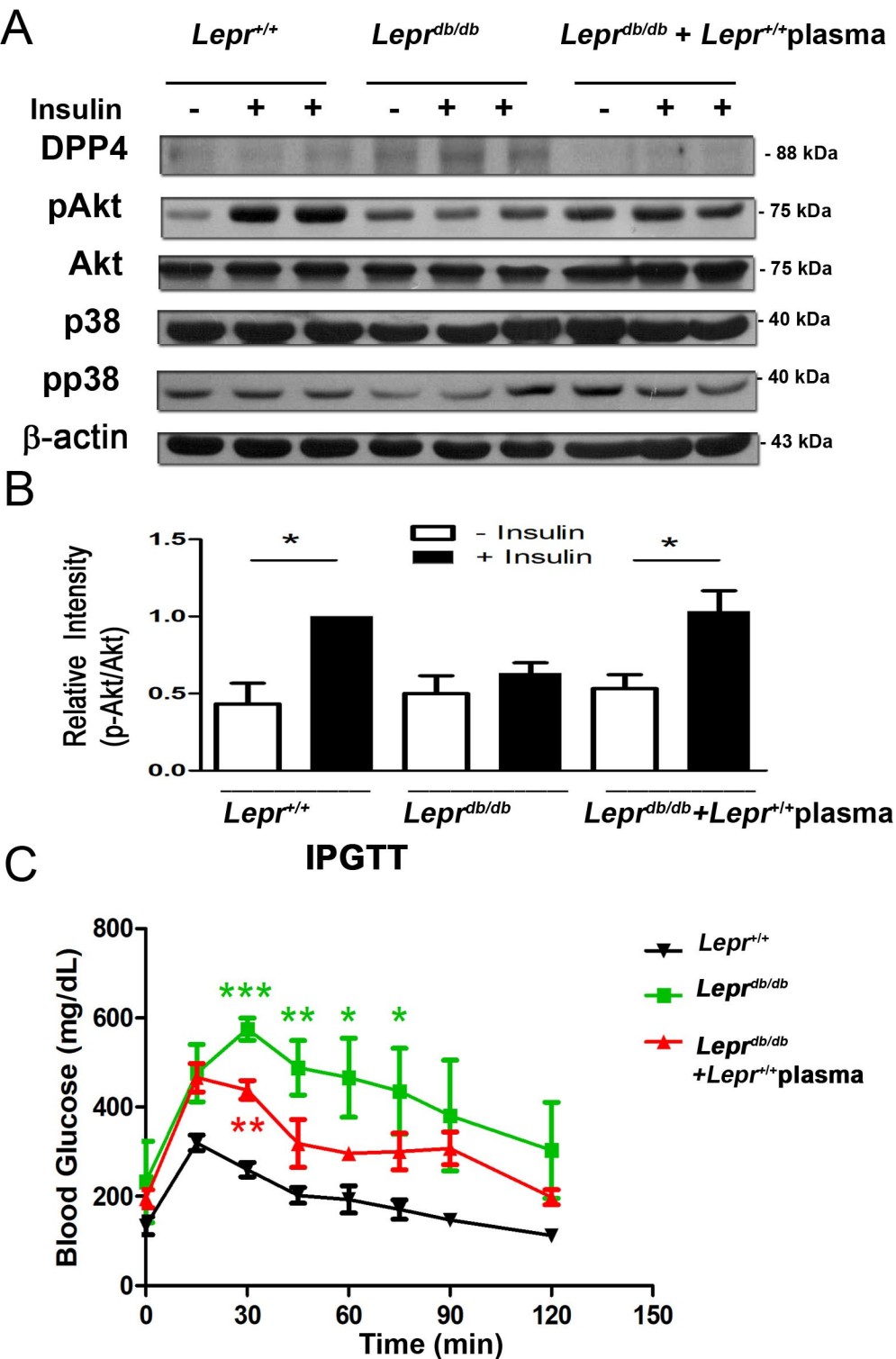

**Fig 7. Lepr<sup>db/db</sup> mice injected Lepr<sup>+/+</sup> plasma exhibit increased insulin-induced Akt phosphorylation and decreased glucose intolerance.** (A and B) $Lepr^{+/+}$, $Lepr^{db/db}$, and $Lepr^{+/+}$ plasma-injected $Lepr^{db/db}$ mice were administered with or without insulin (1.25 IU/kg body weight). After seven days, animals were euthanized, and livers were harvested and subjected to (A) western blot analysis for DPP4, phosphor-Akt, total Akt, phosphor-p38, and total p38 expression. (B) the relative intensity of p-Akt/Akt was quantified. N = 4/group. $^{*}p{<}0.05$. (C) After 15h fasting, $Lepr^{+/+}$, $Lepr^{db/db}$, and $Lepr^{+/+}$ plasma-injected $Lepr^{db/db}$ mice were administered with glucose (1 g/kg body weight).

The blood glucose level was measured before and every 15 minutes up to 2 hours after glucose administration by using a glucose meter. N = 4/group. $^{**}p<0.01$; $^{***}p<0.001$.

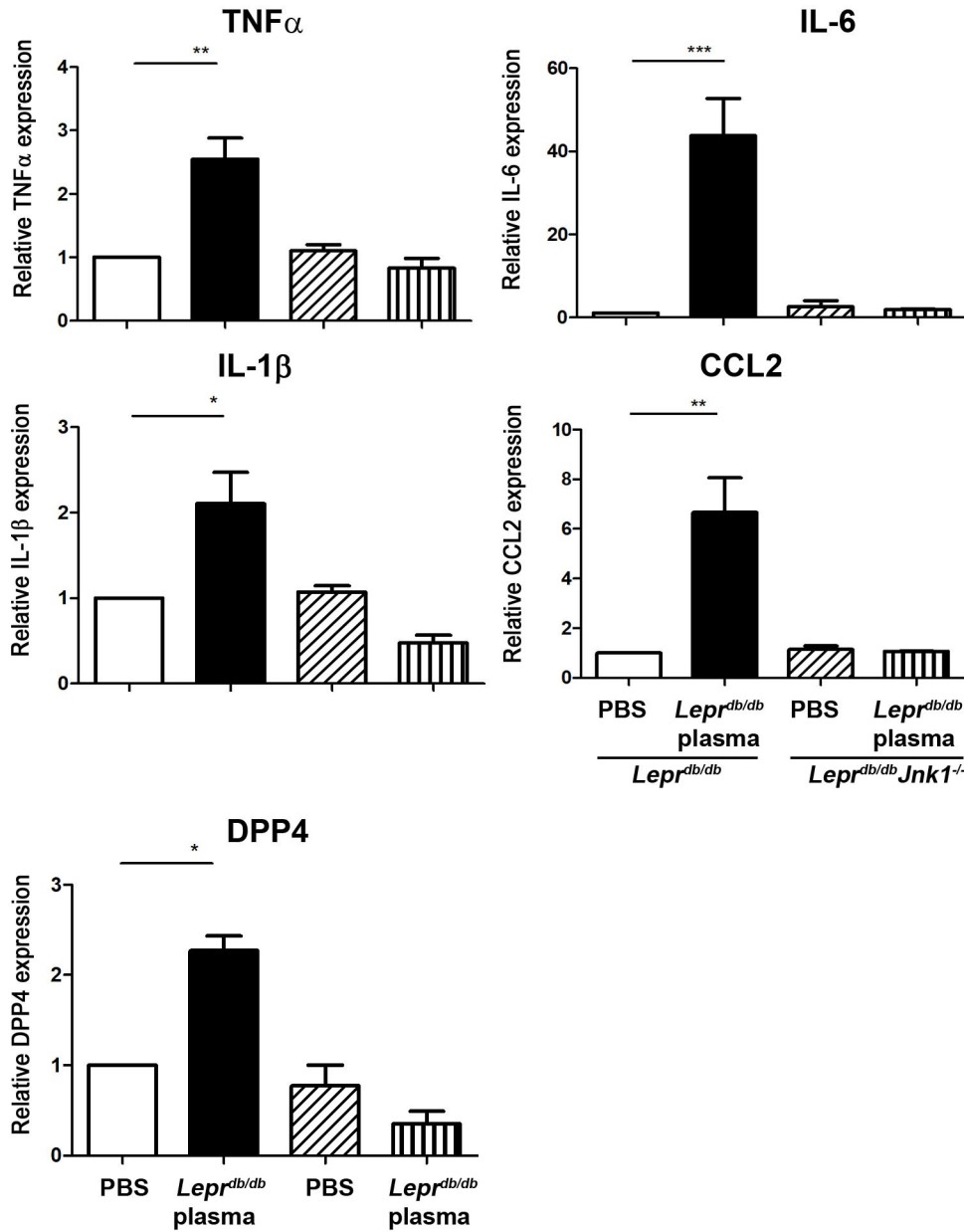

**Fig 8. Diabetic plasma induces M1 expression in adipose tissue from *Lepr<sup>db/db</sup>* mice but not in adipose tissue from *Lepr<sup>db/db</sup>-Jnk1<sup>-/-</sup>* mice.** SVFs were harvested from adipose tissue of *Lepr<sup>db/db</sup>* and *Lepr<sup>db/db</sup>-Jnk1<sup>-/-</sup>* mice. *Lepr<sup>db/db</sup>* and *Lepr<sup>db/db</sup>-Jnk1<sup>-/-</sup>* SVFs were then treated with PBS or 10% *Lepr<sup>db/db</sup>* plasma *in vitro*. Four hours after incubation at 37°C, cells were collected and subjected to Q-PCR analysis for mRNA expression of TNF-α, IL-6, IL-1β, CCL2, and DPP4. N = 5/group. $^{*}p<0.05$; $^{**}p<0.01$; $^{***}p<0.001$.

## Discussion

Adipose tissue dysfunction is characterized by low-grade chronic inflammation in diabetes, metabolic disorders and insulin resistance [17]. Chronic inflammation is one of the major causes for vascular and kidney complications in patients with diabetes. In this study, we demonstrated that M1/M2 cytokine expression in adipose tissue may play an important role in diabetes-induced systemic inflammation and importantly, the reversal of diabetes-induced M1/M2 cytokine expression in the adipose tissue my represent a novel therapeutic strategy for attenuating chronic inflammation-induced systemic complications in diabetes.

M1 macrophages secrete cytokines to activate inflammatory pathways in insulin-target cells, which results in the activation of JNK, κB kinase β inhibitors, and other serine kinases [1]. Our data demonstrated that SVFs from adipose tissue of $Lepr^{db/db}$ mice exhibited enhanced JNK activation compared with those from $Lepr^{+/+}$ mice and injection of plasma from $Lepr^{+/+}$ mice into adipose tissue of $Lepr^{db/db}$ mice suppressed JNK activation compared with those injected with plasma from $Lepr^{db/db}$ mice. Importantly, plasma from $Lepr^{db/db}$ mice induced TNF-α, IL-6, IL-1β, CCL2, and DPP4 mRNA expression in SVFs from $Lepr^{db/db}$ mice but not in SVFs from $Lepr^{db/db}$-$Jnk1^{-/-}$ mice. These results suggest that M1 cytokines activate JNK pathway to promote inflammatory cytokine production and injection of non-diabetic plasma may suppress JNK activation and inflammatory cytokine production in the adipose tissue of diabetes.

DPP4 activity is associated with the onset and severity of obesity and diabetes [13]. The levels of plasma DPP4 activity are elevated in diseases, including type 2 diabetes mellitus (T2DM) [18], obesity [18], and atherosclerosis [19]. Recently, the circulating levels of endogenous soluble DPP4 were found to be dissociated from the extent of systemic and white adipose tissue inflammation [15]. However, we found that diabetes-induced plasma DPP4 activity and DPP4 expression in tissues were closely related with diabetes-induced systemic inflammation and glucose intolerance. First, both female and male $Lepr^{db/db}$ mice demonstrated increased plasma DPP4 activity compared with their corresponding $Lepr^{+/+}$ mice. Second, $Lepr^{db/db}$ mice exhibited increased DPP4 mRNA expression in adipose tissue and liver compared with $Lepr^{+/+}$ mice. Third, in vitro treatment of $Lepr^{db/db}$ SVFs with diabetic plasma significantly induced DPP4 mRNA expression compared with those treated with non-diabetic plasma. However, injection of non-diabetic plasma to adipose tissue of diabetic mice resulted in decreased plasma DPP4 activity and repressed DPP4 expression in adipose tissue and liver in diabetic mice compared to those injected with diabetic plasma. Importantly, injection of non-diabetic plasma into the adipose tissue of diabetic mice attenuated glucose intolerance and insulin resistance in diabetic mice. Altogether, our results demonstrate that diabetes induces adipose tissue and liver DPP4 expression as well as plasma DPP4 activity. In addition, our findings further reveal that the reversal of DPP4 expression in adipose tissue with non-diabetic plasma not only decreases plasma DPP4 activity and liver DPP4 expression but also ameliorates glucose intolerance and insulin resistance in diabetes. Thus our findings suggest that the inhibition of diabetes-induced DPP4 activity could be a plausible therapeutic strategy to reestablish glucose homeostasis in diabetes.

The increased secretions of multiple proinflammatory cytokines and chemokines [20], such as IL-6, TNF-α, IL-1β, and monocyte chemoattractant protein-1 (MCP-1) in adipose tissue are involved in insulin resistance [21]. Ghorpade et al. demonstrated that 10% (vol/vol) plasma from diet-induced obesity mice (DIO) induced higher expression of both MCP1 and IL-6 mRNA expression than 10% plasma from lean mice in SVFs from DIO mice [11]. Furthermore, administration of adipose tissue macrophages from lean mice to obese recipients has been proved to improve glucose tolerance and insulin sensitivity in obese mice through the

secretion of miRNA-containing exosomes [22]. Our data demonstrated that 10% diabetic plasma induced M1 expression, however 10% non-diabetic plasma induced M2 expression in SVFs from diabetic mice. Moreover, injection of non-diabetic plasma reduces M1 cytokine but increases M2 cytokine expression in diabetic adipose tissue. Our data demonstrated that the injection of non-diabetic plasma could be a promising therapeutic strategy to reverse diabetes-induced systemic inflammation and insulin resistance through inhibiting inflammatory cytokines. First, *in vitro* treatment of *Lepr*[db/db] SVFs with *Lepr*[+/+] plasma decreased TNF-α, IL-6, IL-1β, CCL2, and DPP4 but increased IL-10 mRNA expression. Second, injection of non-diabetic plasma into adipose tissue of *Lepr*[db/db] mice decreased TNF-α, IL-6, IL-1β, CCL2, and DPP4 mRNA expression as well as pJNK and DPP4 protein expression while increasing IL-10 mRNA expression in adipose tissue of diabetic mice.

Third, injection of non-diabetic plasma decreased ICAM, FMO3, IL-1β, iNOS, TNF-α, and IL-6 mRNA expression and ICAM and FMO3 protein expression in liver, and suppressed TNF-α, iNOS, and DPP4 mRNA expression in Kupffer cells in diabetic animals. Previously, we have demonstrated that diabetes increases intestinal iNOS expression, NO levels in the portal vein, IL-1β and TNF-α expression of Kupffer cells in the liver. iNOS inhibition by L-NAME reduces NO levels in the portal vein, IL-1β and TNF-α expression of Kupffer cells in the liver [23]. Here, we demonstrated that injection of *Lepr*[+/+] plasma into the adipose tissue of *Lepr*[db/db] mice decreased mRNA expression of proinflammatory cytokines and DPP4 in Kupffer cells of diabetic mice. Finally, injection of non-diabetic plasma decreased plasma DPP4 activity and CCL2 levels, and ameliorated insulin resistance. Taken altogether, these results suggest that non-diabetic plasma injection into diabetic adipose tissue could be used to reverse diabetes-induced systemic inflammation and insulin resistance through the reversal of M1/M2 cytokine expression.

M2 macrophages are thought to facilitate adipose tissue homeostasis and protect against insulin resistance. M2 polarization of adipose tissue macrophages can be promoted by Th2 cytokine stimulation, such as IL-4 and IL-13. IL-6, a classical proinflammatory cytokine induced during different kinds of inflammation was shown to provoke M2 polarization of adipose tissue [24]. Furthermore, studies also identify multiple adipose tissue-specific mediators of M2 polarization, including transcription factors, adipokines, fatty acids, and other immune cells [25]. Nevertheless, studies have suggested that the M1/M2 paradigm is an oversimplification of the macrophage biology observed *in vivo* [26]. Thus, the term "metabolic activation" has been suggested by Kratz *et al.* [26], which is characterized by an increased machinery of lipid catabolism in adipose tissue macrophages. Our data suggest that non-diabetic plasma injection into adipose tissue could be used to reverse diabetes-induced systemic inflammation and insulin resistance through the reversal of M1/M2 cytokine expression. We have examined different pro-inflammatory cytokine levels in the plasma injected mice. Although injection of *Lepr*[+/+] plasma into the adipose tissue of *Lepr*[db/db] mice did not change plasma levels of TNF-α and IL-6 compared with those injected with *Lepr*[db/db] plasma, injection of *Lepr*[+/+] plasma significantly decreased plasma levels of CCL2 in diabetic mice compared with those injected with *Lepr*[db/db] plasma. On the other hand, treatment of *Lepr*[db/db] plasma induced upregulation of inflammatory mediators, TNF-α, IL-6, IL-1β, CCL2, and DPP4, in *Lepr*[db/db] SVFs. However, that effect was not observed in *Lepr*[db/db]-*Jnk1*[-/-] SVFs treated with *Lepr*[db/db] plasma. Thus, we speculate that diabetic plasma induces JNK activation to promote M1 polarization of adipose tissue in diabetes. Whether non-diabetic plasma regulates JNK activation to promote M2 polarization in diabetic adipose tissue would require further investigations.

Our study had several limitations. We used a single injection and found out that the single injection of non-diabetic plasma could reverse M1/M2 cytokine expression, DPP4 activity, inflammatory cytokine expression of liver, and Kupffer cell activation in diabetes. Future

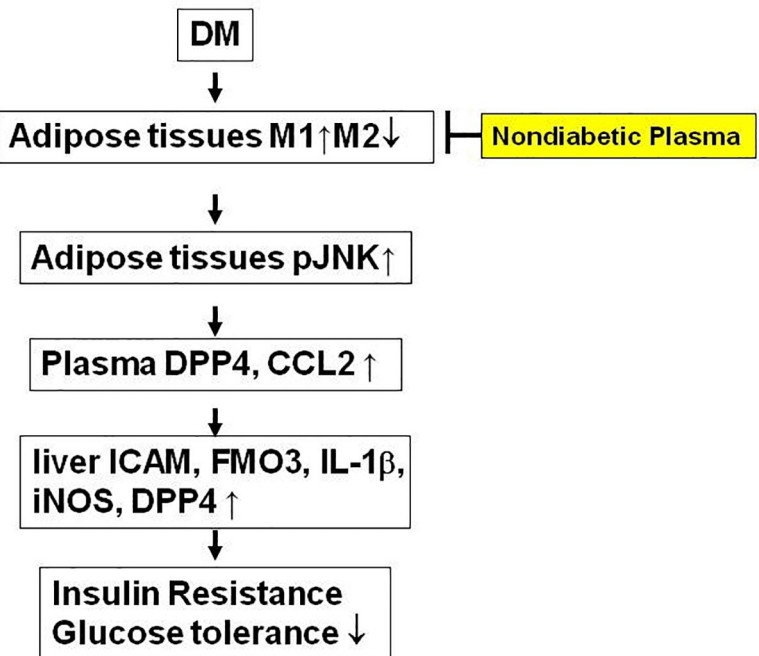

**Fig 9. The model of non-diabetic plasma-mediated suppression of M1 expression and attenuation of insulin resistance and glucose intolerance.** M1 expression in adipose tissue promotes adipose JNK activation, plasma DPP4 and CCL2 induction, and liver ICAM, FMO3, IL-1β, iNOS, and DPP4 upregulation that may result in insulin resistance and impaired glucose tolerance in diabetes. On the other hand, non-diabetic plasma represses diabetic-induced M1 expression in adipose tissue, plasma, and liver that may ultimately attenuate insulin resistance.

studies using multiple injections may have more prominent effect. We did not examine M1/M2 cytokine production in peritoneal macrophages. Also, our data suggest that injection of non-diabetic plasma into adipose tissue could be used to reverse diabetes-induced systemic inflammation and insulin resistance through the reversal of M1/M2 cytokine expression. Whether non-diabetic plasma regulates JNK activation to promote M2 polarization in diabetic adipose tissue would require further investigations.

In summary, we demonstrate the involvement of M1/M2 cytokine expression in diabetes-induced DPP4 activity, insulin resistance, and systemic inflammation (Fig 9). Diabetes induces M1 cytokine expression and decreases M2 cytokine expression in diabetic adipose tissue. Increased adipose tissue M1 expression induces JNK activation, plasma DPP4 activity, and liver ICAM, FMO3, IL-1β, iNOS, TNF-α, IL-6, and DPP4 expression. Subsequently, increased inflammatory cytokine expression and plasma DPP4 activity may promote insulin resistance and glucose intolerance in diabetes. On the other hand, non-diabetic plasma inhibits M1 expression and increases M2 expression in the adipose tissue of diabetes. The inhibition of M1 expression in the adipose tissue represses JNK activation, plasma DPP4 activity and CCL2 levels, as well as liver inflammatory cytokines expression that may ultimately attenuate insulin resistance and glucose intolerance in diabetes.

## Supporting information

**S1 Checklist.**
(DOCX)

**S1 Table. Primer sequences for qPCR.**
(DOC)

**S1 Raw images.**
(PDF)

## Author Contributions

**Conceptualization:** Lee-Wei Chen.

**Funding acquisition:** Lee-Wei Chen.

**Investigation:** Pei-Hsuan Chen.

**Methodology:** Pei-Hsuan Chen.

**Resources:** Jui-Hung Yen.

**Supervision:** Lee-Wei Chen, Jui-Hung Yen.

**Writing – review & editing:** Lee-Wei Chen, Jui-Hung Yen.

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
