## [Decision Letter · Decision Letter 0]

8 Mar 2021

PONE-D-20-35347

Inhibiting adipose tissue M1 cytokine expression decreases DPP4 activity and insulin resistance in a type 2 diabetes mellitus mouse model

PLOS ONE

Dear Dr. Chen,

Thank you for submitting your manuscript to PLOS ONE. After careful consideration, we feel that it has merit but does not fully meet PLOS ONE’s publication criteria as it currently stands. Therefore, we invite you to submit a revised version of the manuscript that addresses the points raised during the review process.

We look forward to receiving your revised manuscript.

Kind regards,

Michael Bader

Academic Editor

PLOS ONE

2. Please note that PLOS does not permit references to “data not shown.” Authors should provide the relevant data within the manuscript, the Supporting Information files, or in a public repository. If the data are not a core part of the research study being presented, we ask that authors remove any references to these data.

3. Please ensure you have discussed any potential limitations of your study in the Discussion.

4. At this time, we request that you  please report additional details in your Methods section regarding animal care, as per our editorial guidelines:

(1) Please state the number of mice used in the study

(2) Please include the method of euthanasia

Thank you for your attention to these requests.

5. Please provide the sequences of the primers used in your PCR experiments.

6. Please provide the  product number and any lot numbers of the antibodies and ELISA kits purchased for your study.

7. PLOS ONE now requires that authors provide the original uncropped and unadjusted images underlying all blot or gel results reported in a submission’s figures or Supporting Information files. This policy and the journal’s other requirements for blot/gel reporting and figure preparation are described in detail at https://journals.plos.org/plosone/s/figures#loc-blot-and-gel-reporting-requirements and https://journals.plos.org/plosone/s/figures#loc-preparing-figures-from-image-files. When you submit your revised manuscript, please ensure that your figures adhere fully to these guidelines and provide the original underlying images for all blot or gel data reported in your submission. See the following link for instructions on providing the original image data: https://journals.plos.org/plosone/s/figures#loc-original-images-for-blots-and-gels.

8. We noticed you have some minor occurrence of overlapping text with the following previous publication(s), which needs to be addressed:

https://academic.oup.com/cardiovascres/article/113/9/1009/3952694

https://www.jimmunol.org/content/198/7/2927

In your revision ensure you cite all your sources (including your own works), and quote or rephrase any duplicated text outside the methods section. Further consideration is dependent on these concerns being addressed.

Reviewers' comments:

Reviewer's Responses to Questions

**Comments to the Author**

1. Is the manuscript technically sound, and do the data support the conclusions?

Reviewer #1: Partly

Reviewer #2: Yes

2. Has the statistical analysis been performed appropriately and rigorously? 

Reviewer #1: Yes

Reviewer #2: Yes

3. Have the authors made all data underlying the findings in their manuscript fully available?

Reviewer #1: Yes

Reviewer #2: Yes

4. Is the manuscript presented in an intelligible fashion and written in standard English?

Reviewer #1: Yes

Reviewer #2: Yes

5. Review Comments to the Author

Reviewer #1: This manuscript intends to evaluate the regulatory mechanisms of M1/M2 polarization in adipose tissue under chronic inflammation and related complications in diabetes with an emphasis on the expression of DPP4.

This is an area of emerging interest as the community is continuing learning on the intersection of inflammation and metabolic disease.

Significant improvements in the manuscript are necessary to support the final model as presented.

Within the introduction, it would increase the focus of the manuscript to include more details on which inflammatory mediators produced by M1 macrophages are implicated for the development of insulin resistance and what is the knowledge of these inflammatory mediators in the context of references 8 and 10 and additionally the authors are missing discussion on the important reference: Zhong et al, A Potential Role for Dendritic Cell/Macrophage-Expressing DPP4 in Obesity-Induced Visceral Inflammation, Diabetes 2013,62(1)149-157

Additionally, the authors demonstrate that SVFs from Lepr db/db mice compared with Lepr +/+ demonstrated a significant increase in TNF-α, IL-6, IL-1β, CCL2, and DPP4 and reduction in IL-10 mRNA expression. It is unclear why these markers were measured and increased rationale in the introduction would add to the manuscript. What about other classic markers of M1/M2?

What are the Kupffer cells characterized as M1/M2?

The authors reproduce data that plasma DPP4 is elevated in both males and females and also that CCL2 is elevated. Are the other factors increased in adipose tissue also elevated? IL-6, IL-1Beta, TNFalpha? How does this influence the model as presented?

The rationale for male Lepr db/db mice to receive injections of 1mL of PBS, 10% plasma from Lepr+/+ or Lepr db/db was not strong. Why 10% plasma injected into adipose tissue? These data are hard to interpret given the glycemia (and many other metabolic factors are elevated only in the Lepr db/db) and need further context and controls to aid in interpretation as discussed. It is unclear which is the dominant factor and how this injection may influence glucose metabolism and inflammation.It is also unclear why differences occur between adipose and liver. Why was 7 days chosen between injection and liver harvest?

For the glucose tolerance test it appears the Lepr db/db mice are at the maximum reading for the glucometer? It may be reasonable to repeat this assay a reduced glucose load.

For the experiments comparing Lepr db/db mice and Lepr db/db Jnk-/- mice – the way the data are analyzed with each PBS treatment acting as the normalization for the treatment it cannot be determined what the basal difference between the two models is. They look identical are they? This would be important in establishing whether JNK signaling regulates DPP4 expression.

Small issues

Line 88 of the introduction Chronic inflammation is one of the major causes of vascular and kidney complications in patients with diabetes needs a reference and further context.

Line 98 “In addition, DPP4 has been shown to exert a direct pro-inflammatory role in different cell types, including lymphocytes, macrophages, and smooth muscle cells “ the authours should add more details into what is meant by pro-inflammatory, particularly with respect tp lymphocytes by those references.

The figure labelling under each plasma experiment is visually confusing and needs improvement particularly for Figure 8.

Band sizes for the Western blots should be provided.

References to genes in the text and figure legends should be lower case and italicized.

The last statement in the introduction “Our findings provide a new insight that the inhibition of M1 cytokine expression in adipose tissue may represent a novel therapeutic approach for attenuating DPP4 activity, systemic inflammation, and insulin resistance in diabetic patients” should be re-evaluated or further discussion is required on how this authors view this paradigm as a therapeutic strategy.

Reviewer #2: In this study, the authors investigated the M1/M2 cytokine and DPP-4 expression in Leprdb/db mice. They demonstrated that SVFs of Leprdb/db mice increased M1 cytokines, and DPP4 mRNA expression but decreased M2 cytokine mRNA expression. Plasma from Leprdb/db mice induced M1 cytokines and DPP4 mRNA expression and plasma from Lepr+/+ mice induced M2 cytokine mRNA expression in SVFs from Leprdb/db mice. Lepr+/+ plasma induced the suppression of M1 cytokines and DPP4 expression, and the suppression of JNK phosphorylation in the SVF of Leprdb/db mice. They also reduced ICAM, FMO3, IL-1β, iNOS, TNF-α, IL-6, and DPP4 expression in liver, and suppressed M1 cytokine and DPP4 in Kupffer cells. Moreover, plasma from Leprdb/db mice did not induce M1 cytokine expression in SVFs from Leprdb/db-JNK1-/- mice. Therefore, they concluded that M1/M2 cytokine expression in adipose tissue is critical in diabetes-induced DPP4 activity, liver inflammation, and insulin resistance.

Specific comments

1. Results section:

Is the change in macrophage number in adipose tissue involved in M1/M2 cytokine expression?

2. Results section:

In ipGTT, since no increase in blood glucose level of 600 mg/dl or more was observed in Leprdb/db mice, accurate data evaluation has not been possible. Although there is no description in the manuscript, it is considered to be a problem of the measurement limit of the blood glucose measuring device. The authors need to show accurate changes in blood glucose levels, for example, changing the measuring equipment. Alternatively, the observation period should be shortened so that the upper limit of measurement is not exceeded.

3. Materials and Methods section:

The methods of In vivo injection of plasma into inguinal adipose tissue is unclear. With this method, it is possible that serum transfer partially to the peritoneum and liver via the portal vein. Authors should show data that this method does not transfer serum directly to the liver. What about M1/M2 cytokine production in peritoneal macrophages?

6. PLOS authors have the option to publish the peer review history of their article (what does this mean?). If published, this will include your full peer review and any attached files.

Reviewer #1: No

Reviewer #2: **Yes: **Takeshi Matsumura

---

## [Author Response · Author response to Decision Letter 0]

24 Apr 2021

PONE-D-20-35347

Ans: We have modified this manuscript to meet PLOS ONE's style requirements.

2. Please note that PLOS does not permit references to “data not shown.” Authors should provide the relevant data within the manuscript, the Supporting Information files, or in a public repository. If the data are not a core part of the research study being presented, we ask that authors remove any references to these data.

Ans: 1. We have removed relative information to the supporting information. 

3. Please ensure you have discussed any potential limitations of your study in the Discussion.

 Ans: We have added the limitations of the study in the discussion section. Our study had several limitations. We used a single injection and found out that the single injection could reverse M1/M2 cytokine expression, DPP4 activity, inflammatory cytokine expression of liver, and Kupffer cell activation in diabetes. Future studies using multiple injections may have more prominent effect. We did not examine M1/M2 cytokine production in peritoneal macrophages. Also, our data suggest that injection of non-diabetic plasma into adipose tissue could be used to reverse diabetes-induced systemic inflammation and insulin resistance through the reversal of M1/M2 cytokine expression. Whether non-diabetic plasma regulates JNK activation to promote M2 polarization in diabetic adipose tissue would require further investigations. 

4. At this time, we request that you please report additional details in your Methods section regarding animal care, as per our editorial guidelines:

 (1) Please state the number of mice used in the study

Ans: The number of mice used in the study has been added. 

(2) Please include the method of euthanasia

Ans: The method of euthanasia has been added.

5. Please provide the sequences of the primers used in your PCR experiments.

Ans: Sequences of the primers have been provided in supported information. 

6. Please provide the product number and any lot numbers of the antibodies and ELISA kits purchased for your study.

Ans: The product number and lot numbers of the antibodies and ELISA kits have been provided. 

7. PLOS ONE now requires that authors provide the original uncropped and unadjusted images underlying all blot or gel results reported in a submission’s figures or Supporting Information files. This policy and the journal’s other requirements for blot/gel reporting and figure preparation are described in detail at https://journals.plos.org/plosone/s/figures#loc-blot-and-gel-reporting-requirements and https://journals.plos.org/plosone/s/figures#loc-preparing-figures-from-image-files. When you submit your revised manuscript, please ensure that your figures adhere fully to these guidelines and provide the original underlying images for all blot or gel data reported in your submission. See the following link for instructions on providing the original image data: https://journals.plos.org/plosone/s/figures#loc-original-images-for-blots-and-gels.

Ans: We have provided the original uncropped and unadjusted images in submission’s figures. 

8. We noticed you have some minor occurrence of overlapping text with the following previous publication(s), which needs to be addressed:

https://academic.oup.com/cardiovascres/article/113/9/1009/3952694

https://www.jimmunol.org/content/198/7/2927

In your revision ensure you cite all your sources (including your own works), and quote or rephrase any duplicated text outside the methods section. Further consideration is dependent on these concerns being addressed.

Ans: The overlapping text has been revised. Duplicated text has been rephrased. 

Reviewers' comments:

Reviewer #1: This manuscript intends to evaluate the regulatory mechanisms of M1/M2 polarization in adipose tissue under chronic inflammation and related complications in diabetes with an emphasis on the expression of DPP4.

This is an area of emerging interest as the community is continuing learning on the intersection of inflammation and metabolic disease.

Significant improvements in the manuscript are necessary to support the final model as presented.

Within the introduction, it would increase the focus of the manuscript to include more details on which inflammatory mediators produced by M1 macrophages are implicated for the development of insulin resistance and what is the knowledge of these inflammatory mediators in the context of references 8 and 10 and additionally the authors are missing discussion on the important reference: Zhong et al, A Potential Role for Dendritic Cell/Macrophage-Expressing DPP4 in Obesity-Induced Visceral Inflammation, Diabetes 2013,62(1)149-157

Ans: 1. We have added the followings in the introduction section: M1 macrophages produce inflammatory cytokines, such as TNF-α, IL-6, and IL-1β, that inhibit the ability of adipocytes to respond to insulin. M1 macrophages secrete cytokines, such as TNF-α and IL-6, to activate inflammatory pathways in insulin target cells, resulting in the activation of Jun N-terminal kinase (JNK) and inhibition of κB kinase β [1].

2. This reference has been added in the introduction section and the following statements have been added. Moreover, DPP4 expression in inflammatory cells such as dendritic cells and macrophages may have a significant role in modulating the adipose tissue inflammation in obesity through its nonenzymatic function [2]. 

Additionally, the authors demonstrate that SVFs from Lepr db/db mice compared with Lepr +/+ demonstrated a significant increase in TNF-α, IL-6, IL-1β, CCL2, and DPP4 and reduction in IL-10 mRNA expression. It is unclear why these markers were measured and increased rationale in the introduction would add to the manuscript. What about other classic markers of M1/M2?

Ans: Different M1/M2 markers could be detected by different methods. We used qPCR to evaluate the M1/M2 markers in this manuscript. Therefore, we have added the references and following paragraph in the introduction section: These M1 macrophages produce inflammatory cytokines, such as TNF-α, IL-6, and IL-1β, that inhibit the ability of adipocytes to respond to insulin. M1 macrophages secrete cytokines, such as TNF-α and IL-6, to activate inflammatory pathways in insulin target cells, resulting in the activation of Jun N-terminal kinase (JNK) and inhibition of κB kinase β [1]. On the other hand, M2 adipose tissue macrophages, which are the major resident macrophages in lean adipose tissue, are characterized by high expression of IL-10 and CD206 [3]. 

What are the Kupffer cells characterized as M1/M2?

Ans: Previously, we have demonstrated that diabetes induces proinflammatory cytokines of Kupffer cells. We have added the following paragraph in the discussion section: Previously, we have demonstrated that diabetes increases intestinal iNOS expression, NO levels in the portal vein, IL-1β and TNF-α expression of Kupffer cells in the liver. iNOS inhibition by L-NAME reduces NO levels in the portal vein, IL-1β and TNF-α expression of Kupffer cells in the liver [4]. Here, we further proved that injection of Lepr+/+ plasma into the adipose tissue of Leprdb/db mice decreased mRNA expression of proinflammatory cytokines and DPP4 in Kupffer cells of diabetic mice.

The authors reproduce data that plasma DPP4 is elevated in both males and females and also that CCL2 is elevated. Are the other factors increased in adipose tissue also elevated? IL-6, IL-1Beta, TNFalpha? How does this influence the model as presented?

Ans: We have examined plasma IL-6, IL-1Beta, TNFalpha levels in Leprdb/db mice as well as in Lepr+/+ mice. There were no significant differences of plasma IL-6, IL-1Beta, TNFalpha levels between Leprdb/db mice and control mice. 

The rationale for male Lepr db/db mice to receive injections of 1mL of PBS, 10% plasma from Lepr+/+ or Lepr db/db was not strong. Why 10% plasma injected into adipose tissue? These data are hard to interpret given the glycemia (and many other metabolic factors are elevated only in the Lepr db/db) and need further context and controls to aid in interpretation as discussed. 

Ans: Ghorpade et al. demonstrated that 10% (vol/vol) plasma from diet-induced obesity mice (DIO) induced higher expression of both MCP1 and IL-6 mRNA expression than 10% plasma from lean mice in SVFs from DIO mice [5]. Our data demonstrated that 10% diabetic plasma induced M1 expression, however 10% non-diabetic plasma induced M2 expression in SVFs of diabetic mice. Moreover, injection of non-diabetic plasma reduces M1 cytokine but increases M2 cytokine expression in diabetic adipose tissue. These description have added in the discussion section. 

It is unclear which is the dominant factor and how this injection may influence glucose metabolism and inflammation. It is also unclear why differences occur between adipose and liver. Why was 7 days chosen between injection and liver harvest?

Ans: 1. Administration of adipose tissue macrophages from lean mice to obese recipients has been proved to improve glucose tolerance and insulin sensitivity in obese mice through the secretion of miRNA-containing exosomes [6]. We demonstrated that non-diabetic plasma inhibits M1 expression and increases M2 expression in the adipose tissue of diabetes. The inhibition of M1 expression in the adipose tissue represses JNK activation, plasma DPP4 activity and CCL2 levels, as well as liver inflammatory cytokines expression that may ultimately attenuate insulin resistance and glucose intolerance in diabetes. 

2. We have added the limitation of this study in the discussion section as followings: We used a single injection and found out that the single injection could reverse M1/M2 cytokine expression, DPP4 activity, inflammatory cytokine expression of liver, and Kupffer cell activation in diabetes. Future studies using multiple injections may have more prominent effect. Also, our data suggest that non-diabetic plasma injection into adipose tissue could be used to reverse diabetes-induced systemic inflammation and insulin resistance through the reversal of M1/M2 cytokine expression. Whether non-diabetic plasma regulates JNK activation to promote M2 polarization in diabetic adipose tissue would require further investigations.

For the glucose tolerance test it appears the Lepr db/db mice are at the maximum reading for the glucometer? It may be reasonable to repeat this assay a reduced glucose load.

Ans: We have repeated the glucose tolerance test and decreased the injection of glucose from 2 g/kg body weight to 1 g/kg body weight. The figure 7C has been revised. 

For the experiments comparing Lepr db/db mice and Lepr db/db Jnk-/- mice – the way the data are analyzed with each PBS treatment acting as the normalization for the treatment it cannot be determined what the basal difference between the two models is. They look identical are they? 

Ans: Thanks for reviewer’s suggestions. We have repeated the experiments Fig 8 and revised the figures.

Small issues

Line 88 of the introduction Chronic inflammation is one of the major causes of vascular and kidney complications in patients with diabetes needs a reference and further context.

Line 98 “In addition, DPP4 has been shown to exert a direct pro-inflammatory role in different cell types, including lymphocytes, macrophages, and smooth muscle cells “ the authours should add more details into what is meant by pro-inflammatory, particularly with respect to lymphocytes by those references.

Ans: 1. We have added the followings in introduction section: Chronic inflammation is one of the major causes of vascular and kidney complications in patients with diabetes [7]. The inflammatory response could be stimulated by various mechanisms, including hyperglycemia-induced cell death, which increases the aggregation of macrophages in the kidney [8].

2. Also, we have added the followings in the introduction section. Haematopoietic cells have been found to be a major source of circulating soluble DPP-4 (sDPP4). Moreover, activation of mouse and human T lymphocytes induced sDPP4 [9].

The figure labelling under each plasma experiment is visually confusing and needs improvement particularly for Figure 8.

Ans: The figure labelling under each plasma experiment has been revised including Figure 8. 

Band sizes for the Western blots should be provided.

Ans: Band sizes for the western blots have been provided.

References to genes in the text and figure legends should be lower case and italicized.

Ans: 1. References to genes in the text and figure legends has been changed to lower cases and italicized.

The last statement in the introduction “Our findings provide a new insight that the inhibition of M1 cytokine expression in adipose tissue may represent a novel therapeutic approach for attenuating DPP4 activity, systemic inflammation, and insulin resistance in diabetic patients” should be re-evaluated or further discussion is required on how this authors view this paradigm as a therapeutic strategy.

Ans: We have changed the last statement in the introduction as followings: Our findings provide a new insight that the inhibition of M1 cytokine expression in adipose tissue with non-diabetic plasma may represent a novel therapeutic approach for attenuating DPP4 activity, systemic inflammation, and insulin resistance in diabetic patients.. 

Reviewer #2: In this study, the authors investigated the M1/M2 cytokine and DPP-4 expression in Leprdb/db mice. They demonstrated that SVFs of Leprdb/db mice increased M1 cytokines, and DPP4 mRNA expression but decreased M2 cytokine mRNA expression. Plasma from Leprdb/db mice induced M1 cytokines and DPP4 mRNA expression and plasma from Lepr+/+ mice induced M2 cytokine mRNA expression in SVFs from Leprdb/db mice. Lepr+/+ plasma induced the suppression of M1 cytokines and DPP4 expression, and the suppression of JNK phosphorylation in the SVF of Leprdb/db mice. They also reduced ICAM, FMO3, IL-1β, iNOS, TNF-α, IL-6, and DPP4 expression in liver, and suppressed M1 cytokine and DPP4 in Kupffer cells. Moreover, plasma from Leprdb/db mice did not induce M1 cytokine expression in SVFs from Leprdb/db-JNK1-/- mice. Therefore, they concluded that M1/M2 cytokine expression in adipose tissue is critical in diabetes-induced DPP4 activity, liver inflammation, and insulin resistance.

Specific comments

1. Results section:

Is the change in macrophage number in adipose tissue involved in M1/M2 cytokine expression?

Ans: We have examined the number of cells of SVFs. There are about 3 x 109 cells/per mice. Different treatments did not significantly change the number of cells in SVFs. However, we did not examine the changes of macrophages in SVFs after treatments. We have added the following in the method section: For in vitro treatment, 1ml of 10% plasma was added to the equal amount of SVFs (1 × 109 cells/ml),

2. Results section:

In ipGTT, since no increase in blood glucose level of 600 mg/dl or more was observed in Leprdb/db mice, accurate data evaluation has not been possible. Although there is no description in the manuscript, it is considered to be a problem of the measurement limit of the blood glucose measuring device. The authors need to show accurate changes in blood glucose levels, for example, changing the measuring equipment. Alternatively, the observation period should be shortened so that the upper limit of measurement is not exceeded.

Ans: Thanks for reviewer’s suggestion. We have repeated the glucose tolerance test and decreased the injection of glucose from 2 g/kg body weight to 1 g/kg body weight. The figure 7C has been revised. 

3. Materials and Methods section:

The methods of In vivo injection of plasma into inguinal adipose tissue is unclear. With this method, it is possible that serum transfer partially to the peritoneum and liver via the portal vein. Authors should show data that this method does not transfer serum directly to the liver. What about M1/M2 cytokine production in peritoneal macrophages?

Ans: 1. Thanks for reviewer’s suggestion. We injected plasma into the middle of the adipose tissue of inguinal area. We did not inject plasma into peritoneum. Plasma of Leprdb/db mice showed high CCL2 levels compared with WT mice. However, injection of plasma of Leprdb/db mice to adipose tissue of Leprdb/db mice did not induce high CCL2 levels in the liver or Kupffer cells. Therefore, it is unlikely that the injected plasma was transferred to the liver via the portal vein. Furthermore liver and Kupffer cells did not show IL-10 expression after the injection of plasma of Lepr+/+ mice. This suggest that effect of Leprdb/db plasma or Lepr+/+ mice on the liver may be through the changes of adipose tissue rather than directly on liver. 

2. We did not examine M1/M2 cytokine production in peritoneal macrophages. This has been added in the limitation of this study.

3. Previously, administration of adipose tissue macrophages from lean mice to obese recipients has been proved to improve glucose tolerance and insulin sensitivity in obese mice through the secretion of miRNA-containing exosomes [6]. Ghorpade et al. demonstrated that 10% (vol/vol) plasma from diet-induced obesity mice (DIO) induced higher expression of both MCP1 and IL-6 mRNA expression than 10% plasma from lean mice in SVFs from DIO mice [5]. We further demonstrated that diabetic plasma induces M1 expression, however non-diabetic plasma induces M2 expression in SVFs from diabetic mice. Moreover, injection of non-diabetic plasma reduces M1 cytokine but increases M2 cytokine expression in diabetic adipose tissue and attenuates insulin resistance and glucose intolerance in diabetes. Therefore, our data suggest that non-diabetic plasma inhibits M1 expression and increases M2 expression in the adipose tissue of diabetes, decreases liver inflammatory cytokines expression, and attenuates insulin resistance and glucose intolerance in diabetes.

References

1. Russo L, Lumeng CN. Properties and functions of adipose tissue macrophages in obesity. Immunology. 2018;155(4):407-17. doi: 10.1111/imm.13002. PubMed PMID: 30229891; PubMed Central PMCID: PMC6230999.

2. Zhong J, Rao X, Deiuliis J, Braunstein Z, Narula V, Hazey J, et al. A potential role for dendritic cell/macrophage-expressing DPP4 in obesity-induced visceral inflammation. Diabetes. 2013;62(1):149-57. doi: 10.2337/db12-0230. PubMed PMID: 22936179; PubMed Central PMCID: PMC3526020.

3. Fujisaka S, Usui I, Bukhari A, Ikutani M, Oya T, Kanatani Y, et al. Regulatory mechanisms for adipose tissue M1 and M2 macrophages in diet-induced obese mice. Diabetes. 2009;58(11):2574-82. doi: 10.2337/db08-1475. PubMed PMID: 19690061; PubMed Central PMCID: PMC2768159.

4. Lin SH, Chung PH, Wu YY, Fung CP, Hsu CM, Chen LW. Inhibition of nitric oxide production reverses diabetes-induced Kupffer cell activation and Klebsiella pneumonia liver translocation. PLoS One. 2017;12(5):e0177269. doi: 10.1371/journal.pone.0177269. PubMed PMID: 28493939; PubMed Central PMCID: PMC5426676.

5. Ghorpade DS, Ozcan L, Zheng Z, Nicoloro SM, Shen Y, Chen E, et al. Hepatocyte-secreted DPP4 in obesity promotes adipose inflammation and insulin resistance. Nature. 2018;555(7698):673-7. doi: 10.1038/nature26138. PubMed PMID: 29562231; PubMed Central PMCID: PMC6021131.

6. Ying W, Riopel M, Bandyopadhyay G, Dong Y, Birmingham A, Seo JB, et al. Adipose Tissue Macrophage-Derived Exosomal miRNAs Can Modulate In Vivo and In Vitro Insulin Sensitivity. Cell. 2017;171(2):372-84 e12. doi: 10.1016/j.cell.2017.08.035. PubMed PMID: 28942920.

7. Dregan A, Charlton J, Chowienczyk P, Gulliford MC. Chronic inflammatory disorders and risk of type 2 diabetes mellitus, coronary heart disease, and stroke: a population-based cohort study. Circulation. 2014;130(10):837-44. doi: 10.1161/CIRCULATIONAHA.114.009990. PubMed PMID: 24970784.

8. Furuya F, Ishii T, Kitamura K. Chronic Inflammation and Progression of Diabetic Kidney Disease. Contributions to nephrology. 2019;198:33-9. doi: 10.1159/000496526. PubMed PMID: 30991405.

9. Casrouge A, Sauer AV, Barreira da Silva R, Tejera-Alhambra M, Sanchez-Ramon S, IcareB, et al. Lymphocytes are a major source of circulating soluble dipeptidyl peptidase 4. Clin Exp Immunol. 2018;194(2):166-79. doi: 10.1111/cei.13163. PubMed PMID: 30251416; PubMed Central PMCID: PMC6194339.

---

## [Decision Letter · Decision Letter 1]

11 May 2021

Inhibiting adipose tissue M1 cytokine expression decreases DPP4 activity and insulin resistance in a type 2 diabetes mellitus mouse model

PONE-D-20-35347R1

Dear Dr. Chen,

We’re pleased to inform you that your manuscript has been judged scientifically suitable for publication and will be formally accepted for publication once it meets all outstanding technical requirements.

Kind regards,

Michael Bader

Academic Editor

PLOS ONE

Additional Editor Comments (optional):

Reviewers' comments:

Reviewer's Responses to Questions

**Comments to the Author**

1. If the authors have adequately addressed your comments raised in a previous round of review and you feel that this manuscript is now acceptable for publication, you may indicate that here to bypass the “Comments to the Author” section, enter your conflict of interest statement in the “Confidential to Editor” section, and submit your "Accept" recommendation.

Reviewer #1: All comments have been addressed

Reviewer #2: All comments have been addressed

2. Is the manuscript technically sound, and do the data support the conclusions?

Reviewer #1: Yes

Reviewer #2: Yes

3. Has the statistical analysis been performed appropriately and rigorously? 

Reviewer #1: Yes

Reviewer #2: Yes

4. Have the authors made all data underlying the findings in their manuscript fully available?

Reviewer #1: Yes

Reviewer #2: Yes

5. Is the manuscript presented in an intelligible fashion and written in standard English?

Reviewer #1: Yes

Reviewer #2: Yes

6. Review Comments to the Author

Reviewer #1: The authors have added additional references and context throughout and repeated experiments to allow for appropriate measurments.

Reviewer #2: In the revised manuscript, the authors have adequately responded to all the comments I raised in the second review.

7. PLOS authors have the option to publish the peer review history of their article (what does this mean?). If published, this will include your full peer review and any attached files.

Reviewer #1: No

Reviewer #2: No

---

## [Editor Report · Acceptance letter]

18 May 2021

PONE-D-20-35347R1 

Inhibiting adipose tissue M1 cytokine expression decreases DPP4 activity and insulin resistance in a type 2 diabetes mellitus mouse model 

Dear Dr. Chen:

I'm pleased to inform you that your manuscript has been deemed suitable for publication in PLOS ONE. Congratulations! Your manuscript is now with our production department. 

Kind regards, 

on behalf of

Prof. Michael Bader 

Academic Editor

PLOS ONE